# QuAFL: Federated Averaging Made Asynchronous and Communication-Efficient

## Abstract

Federated Learning (FL) is an emerging paradigm to enable the large-scale distributed training of machine learning models, while still allowing individual nodes to maintain local data. In this work, we take steps towards addressing two of the main practical challenges when scaling federated optimization to large node counts: the need for tight synchronization between the central authority and individual computing nodes, and the large communication cost of transmissions between the central server and clients. Specifically, we present a new variant of the classic federated averaging (FedAvg) algorithm, which supports both asynchronous communication and communication compression. We provide a new analysis technique showing that, in spite of these system relaxations, our algorithm can provide similar convergence to FedAvg in some parameter regimes. On the experimental side, we show that our algorithm ensures fast convergence for standard federated tasks.

## 1 Introduction

Federated learning (FL) (Konečnỳ et al., 2016; McMahan et al., 2017) is a paradigm for large-scale distributed learning, in which multiple clients, orchestrated by a central authority, cooperate to jointly optimize a machine learning model given their local data. The key promise is to enable joint training over distributed client data, often located on end devices which are computationally- and communication-limited, without the data leaving the client device.

The basic optimization algorithm underlying the learning process is known as *federated averaging (FedAvg)* (McMahan et al., 2017), and works roughly by having a central authority periodically communicate a shared model to all clients; then, the clients optimize this model locally based on their data, and communicate the resulting models to a central authority, which incorporates these models, often via some form of averaging, after which it initiates the next iteration. This algorithmic blueprint has been shown to be effective in practice (Li et al., 2020), and has also motivated a rich line of research analyzing its convergence properties (Stich, 2018; Haddadpour & Mahdavi, 2019), as well as proposing improved variants (Reddi et al., 2020; Karimireddy et al., 2020; Li & Richtárik, 2021).

Scaling federated learning runs into a number of practical challenges (Kairouz et al., 2021). One natural bottleneck is *synchronization* between the server and the clients: as practical deployments may contain thousands of nodes, it is infeasible for the central server to orchestrate synchronous rounds among all participants. A simple mitigating approach is *node sampling*, e.g. (Smith et al., 2017; Bonawitz et al., 2019); another, more general one is *asynchronous communication*, e.g. (Wu et al., 2020; Nguyen et al., 2022b), by which the server and the nodes may work with inconsistent versions of the shared model. An orthogonal scalability barrier is the *high communication cost* of transmitting parameter updates (Kairouz et al., 2021), which may overwhelm communication-limited clients. Several communication-compression approaches have been proposed to address this (Jin et al., 2020; Jhunjhunwala et al., 2021; Li & Richtárik, 2021; Wang et al., 2022).

It is reasonable to assume that *both* these bottlenecks would need to be mitigated in practice: for instance, communication-reduction may not be as effective if the server has to wait for each of the clients to complete their local steps on a version of the model; yet, synchrony is assumed by most references with compressed communication. Yet, removing synchrony completely may lead to divergence, given that local data is usually heterogenous. Thus, it is interesting to ask if asynchrony and communication compression, and heterogenous local data, can be jointly supported.

**Contribution.** In this paper, we address this by proposing an algorithm for **Qu**antized **A**synchronous **F**ederated **L**earning called QuAFL, which is an extension of FedAvg, specifically-adapted to support both asynchronous communication and communication compression. We provide a theoretical analysis of the algorithm's convergence under compressed and asynchronous communication, and experimental results on up to 300 nodes showing that it can also lead to practical performance gains.

**Overview.** The main idea behind QuAFL is that we allow clients to perform their local steps independently of the round structure implemented by the server, and on a local, inconsistent version of the parameters, assuming a probabilistic scheduling model. Specifically, all clients receive a copy of the model when joining the computation, and start performing at most $K \geq 1$ optimization steps on it based on their local data. Independently, in each "logical round," the server samples a set of $s$ clients uniformly at random, and sends them a compressed copy of its current model.

Whenever receiving the server's message, clients *immediately* respond with a compressed version of their *current* model, which may still be in the middle of the local optimization process, and therefore may not include recent server updates, nor the totality of the $K$ local optimization steps. In fact, we even allow that, with some probability, some contacted clients do not take any steps at all. Clients carefully integrate the received server model into their next local iteration, while the server does the same with the client models it receives.

The key missing piece regards quantization. Directly applying standard compressors on transmitted updates (Alistarh et al., 2017b; Karimireddy et al., 2019) runs into the issue that the quantization error may be too large, as it is proportional to the norm of the (updated) model at the client. Resolving this analytically would require either an unrealistic second-moment bound on the maximum gradient update, e.g. (Chen et al., 2021), or variance-reduction techniques (Gorbunov et al., 2021), which may be complex in practice. We circumvent this issue differently, by leveraging a lattice-based quantizer (Davies et al., 2021), which has the property that the quantization error only depends on the *difference* between the quantized model and a carefully-chosen "reference point." We instantiate this technique for the first time in the federated setting.

Our analysis technique relies on a new potential argument, which shows that the discrepancy between the client and server models is always bounded. This bound serves to control the "noise" at different steps due to model inconsistency, but also to ensure that the local models are consistent enough to allow correct encoding and decoding via lattice quantization. The technique is complex yet modular, and should allow further analysis of more complex algorithmic variants.

We validate our algorithm experimentally in the rigorous LEAF (Caldas et al., 2018) environment, on a series of standard tasks. Specifically, in practice, QuAFL can compress updates by more than $3\times$ without significant loss of convergence, and can withstand a large constant fraction of "slow" clients submitting infrequent updates. Moreover, in a setting where client computation speeds are heterogenous, QuAFL provides end-to-end speedup, since the server can progress without waiting for all clients to complete their local computation.

## 2 RELATED WORK

The federated averaging (FedAvg) algorithm was introduced by McMahan et al. (2017), and Stich (2018) was among the first to consider its convergence rate in the homogeneous data setting. Here, we investigate whether one can *jointly* eliminate two of the main scalability bottlenecks of this algorithm, the synchrony between the server and client iterations, as well as the necessity of full-precision communication, with heterogeneous data distributions. Due to space constraints, we focus on prior work which seeks to mitigate these two constraints in the context of FL.

There is significant research into communication-compression for FedAvg (Philippenko & Dieuleveut, 2020; Reisizadeh et al., 2020; Jin et al., 2020; Haddadpour et al., 2021). However, virtually all of this work considers *synchronous* iterations. Reisizadeh et al. (2020) introduced FedPAQ, a variant of FedAvg which supports quantized communication via standard compressors, and provides strong convergence bounds, under the strong assumption of i.i.d. client data. Jin et al. (2020) examines the viability of a variant of the signSGD quantizer (Seide et al., 2014; Karimireddy et al., 2019) in the context of FedAvg, providing convergence guarantees; however, the rate guarantees have a polynomial dependence in the *model dimension* $d$, rendering them less practically meaningful. Haddadpour et al. (2021) proposed FedCOM, a family of federated optimization algorithms with communication-compression and convergence rates; yet, we note that, in order to prove convergence in the challenging heterogeneous-data setting, this reference requires non-trivial technical assumptions on the quantized gradients (Haddadpour et al., 2021, Assumption 5). Chen et al. (2021) also considered update compression, but under convex losses, coupled with a rather strong second-moment bound assumption on the gradients. Finally, Jhunjhunwala et al. (2021) examine adapting the degree of compression during the execution, proving convergence bounds for their scheme, under the non-standard i.i.d. data sampling assumption.

We observe that each of these references requires at least one non-standard assumption for the convergence of FedAvg with compression. By contrast, our analysis works for general (non-convex) losses, under a standard non-i.i.d. data distribution, without relying on second-moment bounds on the gradients.

A complementary approach to reducing communication cost in FL has been to investigate optimizers with faster convergence, e.g. (Mishchenko et al., 2019; Karimireddy et al., 2020), or adaptive optimizers (Reddi et al., 2020; Tong et al., 2020). Tecent work has shown that these approaches can be compatible with communication-compression (Gorbunov et al., 2021; Li & Richtárik, 2021; Wang et al., 2022). Specifically, for non-convex losses, MARINA Gorbunov et al. (2021) offers theoretical guarantees both in terms of convergence and bits transmitted. However, MARINA is structured in synchronous rounds; moreover, it periodically (with some probability) has clients compute full gradients and transmit uncompressed model updates, and requires complex synchronization and variance-reduction to compensate for the extra noise due to quantization. Tyurin & Richtárik (2022) proposed a family of theoretical methods called DASHA, which combines the general structure of MARINA with Momentum Variance Reduction (MVR) methods (Cutkosky & Orabona, 2019), partially relaxing the coupling between the server and the workers and allowing compressed updates.

In contrast to these works, we focus on obtaining a practical algorithm with good convergence bounds: we always transmit compressed, low-precision messages, and consider a notion of asynchronous communication which allows the server and nodes to make progress independently, in non-blocking fashion. We focus on the classic, practical FedAvg algorithm, although our general algorithmic and analytic approach should generalize to more complex notions of local optimization.

Our approach extends ideas from the analysis of decentralized variants of SGD (Lian et al., 2017; Tang et al., 2018; Nadiradze et al., 2021; Koloskova et al., 2019; Lu & De Sa, 2020), bringing them into the context of federated optimization. Significant differences exist: notably, we introduce a novel potential argument, adapted to FL, and cannot rely on stronger assumptions available in the decentralized setting, e.g. a gradient second-moment bound (Lu & De Sa, 2020).

The concurrent work of Koloskova et al. (2022) provided sharper convergence bounds for asynchronous SGD in a model that is related to ours. Specifically, this reference considers a setting with worst-case and average delay bounds on asynchrony, and proves convergence rates that are similar to ours in the case of a single sampled client at a time $s$. By contrast, our work considers a different probabilistic model on the delays, related to that of Cannelli et al. (2020) in which the worst-case delay may be unbounded. In addition, we allow the clients to be interrupted by the server during their local computation, which may lead to practical improvements in terms of waiting times and load-balancing. At the technical level, the two analysis techniques are different: in particular, their technique does not require a lower bound on the number of SGD steps w.r.t. the number of nodes $n$.

## 3 THE ALGORITHM

### 3.1 SYSTEM OVERVIEW

**System Model.** We assume a distributed system with one coordinator and $n$ workers, jointly minimizing a $d$-dimensional, differentiable function $f : \mathbb{R}^d \to \mathbb{R}$. We consider the empirical risk minimization (ERM) setting, in which data samples are located at the $n$ nodes. Each agent $i$ has a local function $f_i$ associated to its own local fraction of the data, i.e $\forall x \in \mathbb{R}^d$: $f(x) = \sum_{i=1}^{n} f_i(x)/n$. The goal is to converge on a model $x^*$ which minimizes the empirical loss. Clients run a distributed variant of SGD, coordinated by the central node. We will assume that each client $i$ is able to obtain *unbiased stochastic gradients* $\widetilde{g}_i$ of its own local function $f_i$, i.e. $\mathbb{E}[\widetilde{g}_i(x)] = \nabla f_i(x)$. These stochastic gradients can be computed by each agent by sampling i.i.d. from its own local distribution. Our analysis will consider the case where each client distribution is distinct, but there is a bound on the maximum gradient discrepancy.

We model *client asynchrony* as follows: between two consecutive interactions with the server, each client should perform a number of gradient steps on its local model. We treat the number of local steps at client $i$ as a random variable $\mathcal{H}_i$, taking values in $\{0, 1, 2, \ldots, K\}$, where $K$ is a bound on how many steps a client can take in isolation. We emphasize the fact that $\mathcal{H}_i$ *can take the value* 0, meaning that the client may take no steps since last contacted. Our only assumption regarding asynchrony is that the expected value of $\mathcal{H}$, denoted by $H$, exists and is $> 0$. That is, we assume that, on average, each client makes non-zero progress, and clients progress at similar rates, although the individual step distributions $\mathcal{H}_i$ can be completely different.

### 3.2 ALGORITHM DESCRIPTION

**Overview.** Our algorithm starts from the standard pattern used by federated averaging (FedAvg): computation and communication are organized in logical "rounds," where in each round the server transmits its current version of the model to either all, or a subset of clients. The clients should then take some number of local optimization steps on the received model, which is at most $K \geq 1$, and transmit the result to the server, which integrates these updates. Our algorithm will relax this pattern in two orthogonal ways, allowing for both quantized and asynchronous communication.

**Quantized Communication.** The first relaxation is to only allow for compressed communication of the server model and of the client updates, via quantization. For this, we employ a carefully-parametrized version of the lattice-based quantization technique of Davies et al. (2021), whose analytical properties we describe in the analysis section. For practical purposes, this quantization technique presents an encoding function $Enc(A)$, which encodes an arbitrary input $A$ to its quantized representation. (We always communicate vectors via their quantized representations.) To "read" an encoded message $Enc(A)$, a node must call the symmetric $Dec(B, Enc(A))$ function, which allows for the "decoding" of the input $Enc(A)$ with respect to a reference point $B$, returning a quantized output $Q(A)$. We formally specify the properties of the compression process in Section 4.

---

**Algorithm 1** Pseudocode for QuAFL Algorithm.

---

% Initial models $X_0 = X^1 = X^2 = ... = X^n = 0^d$, number of local steps $K$
% Encoding ($Enc(A)$) and decoding ($Dec(B, Enc(A))$) functions, with common parametrization.
% **At the Server**:
  1: **for** $t = 0$ **to** $T - 1$ **do**
  2:     Server chooses $s$ clients uniformly at random, let $S$ be the resulting set.
  3:     **for all** clients $i \in S$ **do**
  4:         Server sends $Enc(X_t)$ to the client $i$.
  5:         Server receives $Enc(Y^i)$ from client $i$
  6:         % $Y^i = X^i - \eta \widetilde{h}_i$ is the client's progress since last contacted
  7:         $Q(Y^i) \leftarrow Dec(X_t, Enc(Y^i))$ % Decodes client messages relative to $X_t$
  8:     **end for**
  9:     $X_{t+1} = \frac{1}{s+1} X_t + \frac{1}{s+1} \sum_{i \in S} Q(Y^i)$
 10: **end for**

% **At Client** $i$:
% Upon (asynchronous) contact from the server run INTERACTWITHSERVER
% **Local variables:**
% $X^i$ stores the base client model, following the last server interaction. Initially $0^d$.
% $\widetilde{h}_i$ accumulates local gradient steps since last server interaction, initially $0^d$.
  1: **function** INTERACTWITHSERVER
  2:     $MSG_i \leftarrow Enc(X^i - \eta \widetilde{h}_i)$ % Client $i$ compresses its local progress since last contacted.
  3:     Client sends $MSG_i$ to the server.
  4:     Client receives $Enc(X_t)$ from the server, where $t$ is the current server time.
  5:     $Q(X_t) \leftarrow Dec(X^i, Enc(X_t))$ % Client decodes the message *using its old model* as reference point.
  6:     % The client then updates its local model
  7:     $X^i = \frac{1}{s+1} Q(X_t) + \frac{s}{s+1}(X^i - \eta \widetilde{h}_i)$
  8:     %Finally, it performs $K$ new local steps on the updated $X^i$, unless interrupted again.
  9:     LOCALUPDATES($X^i, K$)
 10:     WAIT( )
 11: **end function**
  1: **function** LOCALUPDATES($X^i, K$)
  2:     $\widetilde{h}_i = 0$ % local gradient accumulator
  3:     **for** $q = 0$ **to** $K - 1$ **do**
  4:         $\widetilde{h}_i^q = \widetilde{g}_i(X^i - \eta \sum_{\ell=0}^{q-1} \widetilde{h}_i^\ell)$ % compute the $q$th local gradient
  5:         $\widetilde{h}_i = \widetilde{h}_i + \widetilde{h}_i^q$ % add it to the accumulator
  6:     **end for**
  7: **end function**

---

**Asynchronous Communication.** A key practical limitation of the FedAvg pattern is that the server and its workers have to communicate in synchronous, lock-step fashion: thus, the server must wait for the results of computation at a round before it can move to the next round. In particular, this means that the server has to wait for *the slowest client* to complete its local steps before it can proceed.

QuAFL relaxes this requirement by essentially allowing any contacted node $i$ to immediately return (a quantized version of) its current version of the model to the server upon being contacted, even though the client might still not have completed all its $K$ local optimization steps for the round. More precisely, the client always records its "base" model at the end of the last interaction with the server into parameter $X^i$, and sums up its gradient updates since the last interaction into the buffer $\widetilde{h}_i$. Upon being contacted, the client simply sends its current progress $Y^i = X^i - \eta\widetilde{h}_i$ to the server (excluding the local step for which computation was not finished due to interruption from the server), where $\eta$ is the learning rate, in quantized form. It is possible that this progress is *zero*. The client then decodes the quantized server model $X_t$, using its old local model $X^i$ as the decoding key. Finally, the client updates $X^i$ to include the server's new information via weighted averaging. It is then ready to restart its local update loop, upon this new model.

It is important to notice that the server interaction occurs *asynchronously*, and that it might occur either while the client is still performing local steps, or after the client has completed its $K$ local steps, and is idle, waiting for server contact. In the former case, upon being contacted, immediately calls the server interaction function, without performing additional steps. (In particular, we allow the number of completed local steps to be $0$.) Globally, the server contacts $s$ random agents in each logical round, sends them a quantized version of the global model $X_t$, then receives quantized versions of their progress, and then incorporates this into the global model which will be sent at the next round.

**Discussion.** The practical advantage of QuAFL is that the server does not have to wait for each of the contacted clients to complete their local optimization on the global model $X_t$. In addition, an important departure from FedAvg is the averaging between the server and client models. Our formulation is important for fast convergence: as we show in Figure 4, other forms, such as just adopting the client average, lead to worse convergence.

## 4 CONVERGENCE ANALYSIS

### 4.1 ANALYTICAL ASSUMPTIONS

We begin by stating the assumptions we make in the theoretical analysis of our algorithm. Specifically, we assume the following for the global loss function $f$, the individual client losses $f_i$, and the stochastic gradients $\widetilde{g}_i$:

1. **Uniform Lower Bound:** There exists $f_* \in \mathbb{R}$ such that $f(x) \geq f_*$ for all $x \in \mathbb{R}^d$.
2. **Smooth Gradients**: For any client $i$, the gradient $\nabla f_i(x)$ is $L$-Lipschitz continuous for some $L > 0$, i.e. for all $x, y \in \mathbb{R}^d$:
$$\|\nabla f_i(x) - \nabla f_i(y)\| \leq L\|x - y\|. \tag{1}$$
3. **Bounded Variance**: For any client $i$, the variance of the stochastic gradients is bounded by some $\sigma^2 > 0$, i.e. for all $x \in \mathbb{R}^d$:
$$\mathbb{E}\left\|\widetilde{g}_i(x) - \nabla f_i(x)\right\|^2 \leq \sigma^2. \tag{2}$$
4. **Bounded Dissimilarity**: There exist constants $G^2 \geq 0$ and $B^2 \geq 1$, s.t. $\forall x \in \mathbb{R}^d$:
$$\sum_{i=1}^n \frac{\left\|\nabla f_i(x)\right\|^2}{n} \leq G^2 + B^2 \left\|\nabla f(x)\right\|^2. \tag{3}$$

The first three conditions are universal in distributed non-convex stochastic optimization, whereas the fourth encodes the fact that there must be a bound on the amount of divergence between the local distributions at the nodes in order to allow for joint optimization (Karimireddy et al., 2020; Jin et al., 2020; Gorbunov et al., 2021).

In addition, we make the following assumption on the local progress performed by each node:

5. **Probabilistic Progress:** The expected number of local steps taken by a client when contacted by the server is $H > 0$.

**Quantization Procedure.** Please recall the semantics of our quantization procedure from Section 3.2. In this context, the quantizer has the following guarantees (Davies et al., 2021) (Lemma 23):

**Lemma 4.1.** *(Lattice Quantization) Fix parameters $R$ and $\gamma > 0$. There exists a quantization procedure defined by an encoding function $Enc_{R,\gamma} : \mathbb{R}^d \to \{0, 1\}^*$ and a decoding function $Dec_{R,\gamma} = \mathbb{R}^d \times \{0, 1\}^* \to \mathbb{R}^d$ such that, for any vector $x \in \mathbb{R}^d$ which we are trying to quantize, and*

*any vector $y$ which is used by decoding, which we call the* decoding key, *if $\|x - y\| \leq R^{R^d}\gamma$ then with probability at least $1 - \log\log(\frac{\|x-y\|}{\gamma})O(R^{-d})$, the function $Q_{R,\gamma}(x) = Dec_{R,\gamma}(y, Enc_{R,\gamma}(x))$ has the following properties:*

1. *(Unbiased decoding)* $\mathbb{E}[Q_{R,\gamma}(x)] = \mathbb{E}[Dec_{R,\gamma}(y, Enc_{R,\gamma}(x))] = x$;

2. *(Error bound)* $\|Q_{R,\gamma}(x) - x\| \leq (R^2 + 7)\gamma$;

3. *(Communication bound)* $O\left(d\log(\frac{R}{\gamma}\|x - y\|)\right)$ *bits are needed to send* $Enc_{R,\gamma}(x)$.

## 4.2 MAIN RESULTS

Roughly, our aim is to show that in our algorithm, local models of the clients stay close to the local model of the server, so that models are consistent, and we can successfully apply the quantizer. Let $\mu_t = (X_t + \sum_{i=1}^{n} X^i)/(n + 1)$ be the mean over all the node models in the system at a given $t$. Our main result shows the following:

**Theorem 4.2.** *Assume the total number of server steps $T \geq \Omega(n^3)$, the learning rate $\eta = \frac{n+1}{sH\sqrt{T}}$, and quantization parameters $R = 2 + T^{\frac{3}{d}}$ and $\gamma^2 = \frac{\eta^2}{(R^2+7)^2}\left(\sigma^2 + 2KG^2 + \frac{f(\mu_0)-f_*}{L}\right)$. Let $H > 0$ be the expected number of local steps already performed by a client when interacting with the server. Then, with probability at least $1 - O(\frac{1}{T})$ we have that Algorithm 1 converges at the following rate*

$$\frac{1}{T}\sum_{t=0}^{T-1}\mathbb{E}\|\nabla f(\mu_t)\|^2 \leq \frac{5(f(\mu_0) - f_*)}{\sqrt{T}} + \frac{8KL(\sigma^2 + 2KG^2)}{H^2\sqrt{T}} + O\left(\frac{n^3KL^2(\sigma^2 + 2KG^2)}{sH^3T}\right)$$

*and uses $O\left(sT(d\log n + \log T)\right)$ expected communication bits in total.*

**Discussion.** The result provides a trade-off between the convergence speed of the algorithm, the variance of the local distributions (given by $\sigma$ and $G$), the sampling set size $s$, and the average number of local steps $H$ performed by a node when contacted by the server.

For constant $s$, $H$ and $K$, this bound appears to be asymptotically-optimal. Specifically, the third term contains similar "nuisance factors" as the second term, with the addition of the $n^3$ factor, and also bounds the extra variance. Crucially, this larger term is divided by $T$, as opposed to $\sqrt{T}$; since $T$ is our asymptotic parameter, it is common to assume that this extra term becomes negligible as $T$ is large, e.g. (Lu & De Sa, 2020).

However, for super-constant $s = \omega(n)$, there is no tangible benefit due to sampling over $s$ clients. Intuitively, this is because of asynchrony: each client gets sampled on average every $n/s$ steps, and therefore will work on a "stale" copy of the model that is $n/s$ rounds old, which affects convergence speed. While this is a limitation of the analysis, from the practical perspective, this can be addressed by observing that, due to asynchrony, in terms of wall-clock time, we can think of the $s$ interaction steps between the server and the clients as happening in parallel. Thus, again, in terms of wall-clock-time, it is reasonable to perform the substitution $T \to sT$ in the above rate calculation, which indeed suggests that our algorithm is able to obtain speedup with respect to the number of sampled clients $s$.

In practice, it should be reasonable to assume that $H = \Theta(K)$, that is, that on average each client $i$ will have completed its local steps on the old version of the model $X^i$ when being contacted: otherwise, the sampling frequency of the server is too high, and prevents clients from making progress on their local optimization, and the server should simply decrease it.

**Convergence at the Server.** Finally, we show that not only convergence at the server, as opposed to the convergence of the mean of the local models as in Theorem 4.2. We get that:

**Corollary 4.3.** *Assume the total number of steps $T \geq \Omega(n^4)$, the learning rate $\eta = \frac{n+1}{sH\sqrt{T}}$, and quantization parameters $R = 2 + T^{\frac{3}{d}}$ and $\gamma^2 = \frac{\eta^2}{(R^2+7)^2}\left(\sigma^2 + 2KG^2 + \frac{f(\mu_0)-f_*}{L}\right)$. Let $H > 0$ be the expected number of local steps already performed by a client when interacting with the server. Then, with probability at least $1 - O(\frac{1}{T})$ we have that Algorithm 1 converges at the following rate*

$$\frac{1}{T}\sum_{t=0}^{T-1}\mathbb{E}\|\nabla f(X_t)\|^2 \leq \frac{5(f(\mu_0) - f_*)}{\sqrt{T}} + \frac{8KL(\sigma^2 + 2KG^2)}{H^2\sqrt{T}} + O\left(\frac{n^4KL^2(\sigma^2 + 2KG^2)}{sH^2T}\right).$$

This corollary yields a very similar bound to our main result, except for the larger dependency between $T$ and $n$, which is intuitively required due to the additional time required for the server to converge

to a similar bound to the mean $\mu_t$. The third term may be significant for large number of nodes $n$; however, since it is divided by $T$ (as opposed to $\sqrt{T}$) it can be seen as negligible for moderate $n$ and large $T$. The fact that QuAFL can match some of the best known rates for FedAvg under some parameter settings may seem surprising, since our algorithm is asynchronous (in particular, nodes take steps on local, delayed versions of the server model) and also supports communication-compression.

## 4.3 Overview of the Analysis

The complete analysis is fairly complex, and is provided in full in the Appendix. Due to space constraints, we only provide an overview of the proofs, outlining the main intermediate results. The first step in the proof is bounding the deviation between the local models and their mean. For this, we introduce the potential function $\Phi_t = \|X_t - \mu_t\|^2 + \sum_{i=1}^n \|X^i - \mu_t\|^2$, and we use a load-balancing approach to show that this potential has the following supermartingale-type property:

**Lemma 4.4.** *For any time step $t$ we have:*

$$\mathbb{E}[\Phi_{t+1}] \leq \left(1 - \frac{1}{4n}\right)\mathbb{E}[\Phi_t] + 8s\eta^2 \sum_{i=1}^n \mathbb{E}\|\widetilde{h}_i\|^2 + 16n(R^2 + 7)^2\gamma^2.$$

The intuition behind this result is that potential $\Phi_t$ will stay well-concentrated around its mean, except for influences from the variance due to local steps (second term) or quantization (third term). With this in place, the next lemma allows us to track the evolution of the average of the local models, with respect to local step and quantization variance:

**Lemma 4.5.** *For any step $t$*

$$\mathbb{E}\|\mu_{t+1} - \mu_t\|^2 \leq \frac{2s^2\eta^2}{n(n+1)^2} \sum_i \mathbb{E}\left\|\widetilde{h}_i\right\|^2 + \frac{2}{(n+1)^2}(R^2 + 7)^2\gamma^2.$$

In both cases, the upper bound depends on the second moment of the nodes' local progress $\sum_i \mathbb{E}\left\|\widetilde{h}_i\right\|^2$. (This is due to the fact that the server contacts $s$ clients, which are chosen uniformly at random.) Then, our main technical lemma uses properties (1), (2) and (3), to concentrate $\sum_i \mathbb{E}\left\|\widetilde{h}_i(X_t^i)\right\|^2$ around the true gradient $\mathbb{E}\|\nabla f(\mu_t)\|^2$, where the expectation is taken over the algorithm's randomness.

**Lemma 4.6.** *For any step $t$, we have that*

$$\sum_{i=1}^n \mathbb{E}\|\widetilde{h}_i\|^2 \leq 2nK(\sigma^2 + 2KG^2) + 8L^2K^2\mathbb{E}[\Phi_t] + 4nK^2B^2\mathbb{E}\|\nabla f(\mu_t)\|^2.$$

We can then combine Lemmas 4.4 and 4.6 to get an upper bound on the potential with respect to $\mathbb{E}\|\nabla f(\mu_t)\|^2$. Summing over steps, we obtain the following:

**Lemma 4.7.**
$$\sum_{t=0}^T \mathbb{E}[\Phi_t] \leq 80Tn^2(R^2+7)^2\gamma^2 + 80Tn^2sK\eta^2(\sigma^2 + 2KG^2) + 160B^2n^2sK^2\eta^2 \sum_{t=0}^{T-1} \mathbb{E}\|\nabla f(\mu_t)\|^2.$$

Next, using the $L$-smoothness of the function $f$, implied by (1), we can show that

$$\mathbb{E}[f(\mu_{t+1})] \leq \mathbb{E}[f(\mu_t)] + \mathbb{E}\langle\nabla f(\mu_t), \mu_{t+1} - \mu_t\rangle + \frac{L}{2}\mathbb{E}\|\mu_{t+1} - \mu_t\|^2. \tag{4}$$

**Final argument.** Using the above inequality, and given that $\mathbb{E}[\mu_{t+1} - \mu_t] = -\frac{\eta}{n+1}\sum_{i\in S}\widetilde{h}_i(X_t^i)$, we observe that the sum $\sum_{i=1}^n \mathbb{E}\langle\nabla f(\mu_t), \mu_{t+1} - \mu_t\rangle$ can be concentrated around $\mathbb{E}\|\nabla f(\mu_t)\|^2$, in similar fashion as in Lemma 4.6. Together with Lemma 4.5, this results in the following bound:

$$\mathbb{E}[f(\mu_{t+1})] - \mathbb{E}[f(\mu_t)] \leq \frac{5\eta sKL^2\mathbb{E}[\Phi_t]}{n(n+1)} + \left(\frac{4sL^2\eta^3K^3}{n+1} + \frac{2s^2K\eta^2L}{(n+1)^2}\right)(\sigma^2 + 2KG^2)$$

$$+ \frac{(R^2+7)^2\gamma^2L}{(n+1)^2} + \left(\frac{-3\eta sH}{4(n+1)} + \frac{8B^2L^2\eta^3sK^3}{n+1} + \frac{4B^2s^2K^2L\eta^2}{(n+1)^2}\right)\mathbb{E}\|\nabla f(\mu_t)\|^2.$$

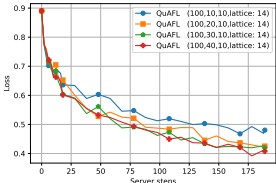

**Figure 1:** Peers $s \in \{10, 20, 30, 40\}$ on convergence, for $n = 100$ clients, 14-bit quantization, on CelebA.

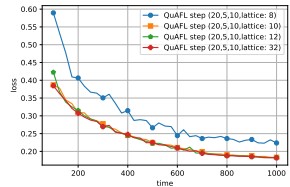

**Figure 2:** Impact of the number of bits $b \in \{8, 10, 12, 32\}$ on convergence, for $n = 40$ clients, $s = 5$ peers on MNIST.

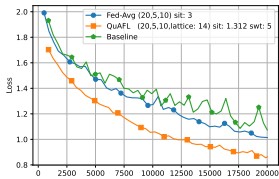

**Figure 3:** Convergence comparison relative to simulated time between QuAFL and FedAvg, for ResNet20/CIFAR10.

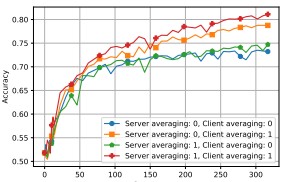

**Figure 4:** Impact of averaging variants on validation accuracy on CelebA, vs. rounds.

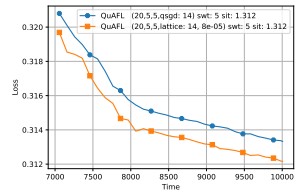

**Figure 5:** Using QSGD vs Lattice quantization in QuAFL, on MNIST.

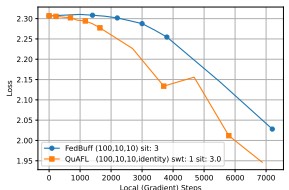

**Figure 6:** Convergence of QuAFL vs. SOTA asynchronous FL algorithm FedBuff.

For $\eta = (n + 1)/\sqrt{T}$, as stated in the Theorem, we can use Lemma 4.7 to cancel out the terms containing the potential $\Phi_t$ (after summing up the inequality over $T$ steps). Replacing these terms, and modulo some additional term wrangling, we obtain the claimed convergence bound.

**Quantization Impact.** Finally, we address the correctness of the quantization technique. We show that the quantization fails with negligible probability:

**Lemma 4.8.** *Let $T \geq \Omega(n^3)$, then for quantization parameters $R = 2 + T^{\frac{3}{d}}$ and $\gamma^2 = \frac{\eta^2}{(R^2+7)^2}(\sigma^2 + 2KG^2 + \frac{f(\mu_0) - f_*}{L})$ we have that the probability of quantization never failing during the entire run of the Algorithm 1 is at least $1 - O\left(\frac{1}{T}\right)$.*

Per Lemma 4.1, in order for the communication to fail with negligible probability, we need to show that whenever the server communicates with a client, the two norm of their local models is at most $R^{R^d}\gamma$. Hence, we need to use bound $\mathbb{E}[\Phi(t)]$. The only similar use of this technique was in Nadiradze et al. (2021); however, the authors of this reference could benefit from assuming that the second-moment of the gradients was bounded. Since we make no such assumption here, we need to find a way to bound $\sum_{t=0}^{T-1} \mathbb{E}\|\nabla f(\mu_t)\|^2$. Fortunately, our main result shows that the gradients are vanishing, so we can take the advantage of the convergence rate and plug it back into Lemma 4.7.

Similarly, due to the Property 3, Lemma 4.1, the number of the bits used by our algorithm, in one communication between the server and a client, depends on the two norm of the distance between their local models. Thus, we can use the bound on $\sum_{t=0}^{T-1} \mathbb{E}\|\nabla f(\mu_t)\|^2$ to show the following.

**Lemma 4.9.** *Let $T \geq \Omega(n^3)$, then for quantization parameters $R = 2 + T^{\frac{3}{d}}$ and $\gamma^2 = \frac{\eta^2}{(R^2+7)^2}(\sigma^2 + 2KG^2 + \frac{f(\mu_0) - f_*}{L})$ we have that the expected number of bits used by Algorithm 1 in total is $O(sT(d\log(n) + \log(T))$.*

We note that the communication cost per step is also asymptotically optimal, modulo the multiplicative $\log n$ and additive $\log T$ terms, required to ensure error probability $1 - O(1/T)$.

## 5 EXPERIMENTAL RESULTS

**Experimental Setup and Goals.** We implemented our algorithm in Pytorch in order to train neural networks for image classification tasks, specifically residual CNNs (He et al., 2016) on the MNIST (LeCun & Cortes, 2010), Fashion MNIST (Xiao et al., 2017), CIFAR-10 (Krizhevsky & Hinton, 2009) and CelebA (Liu et al., 2015) datasets, in the rigorous FL setup of LEAF (Caldas et al., 2018). Details are presented in the Appendix. We aim to validate our analysis relative to the impact of various parameters. We omit error bars, as we observed that the variance is very low.

Specifically, the parameters we examine are $n$, $s$, and $K$, which have the same meaning as in our theoretical analysis. Our experiments are described by $(n, s, K, b)$, where $b$ is the number of bits for quantization. In addition, we define $swt$ as the server waiting time between two consecutive

calls, and the server interaction time, $sit$, as the amount of time that server needs to send and receive necessary data. We assume a *server* and $n$ *clients*. The training dataset is distributed among clients so that each has access to a fixed $1/n$ partition of the training data. We track the accuracy of the server's model on an unseen validation dataset. We measure *loss* and *accuracy* of the model with respect to *simulation time* and *total gradient steps performed by clients*. In each round, the server chooses $s$ clients uniformly at random. It then sends its model to those clients and receives their current local models. Each client will have taken a maximum of $K$ local steps by the time it is contacted by the server. We update the both client and server models following QuAFL, and then increase the server time by $sit$. The server then waits for another interval of *server waiting time (swt)* to make its next call. Unless otherwise stated, communication is compressed.

We differentiate between two types of timing experiments: *uniform timing* experiments assume all clients take the same amount of time for a gradient step; *non-uniform* timing experiments differentiate clients to be *fast* or *slow*. Specifically, the length of each client step is taken to be a random variable $X \sim exp(\lambda)$, where $\lambda$ is $1/2$ for *fast clients* and $1/8$ for *slow* clients; the expected runtime $\mathbb{E}(X)$ would be $2$ and $8$, respectively. In each experiment, we assumed 30% of clients to be *slow*.

Figure 1 examines the impact of the number of sampled peers $s$ when training ResNet18 on the CelebA dataset, where 30% of clients are slow. We first observe that convergence speed clearly follows the ordering of the number of peers $s$, confirming our analysis. Interestingly, timings in this experiment are set up so that there is a 27% probability that a slow client *will not have taken any steps* when interacting with the server. (This probability decreases as $s$ increases.). Thus, this experiment also shows that QuAFL is indeed robust to such slow clients, although their proportion can impact convergence. Figure 2 examines the impact of the number of quantization bits $b$, showing that increasing $b$ from 8 to 10 improves convergence; however, there is clear staturation after 10 bits.

In Figure 3 we examine the loss convergence of FedAvg and QuAFL versus *simulated execution time*, in a system with 20 clients, out of which $25\%$ are slow. (The *Baseline* is a single slow node that performs an optimization step per round.) Here, it is evident that QuAFL asynchrony allows it to provide a faster convergence in terms of wall-clock time, than its synchronous counterparts.

In Figure 4, we examine the impact of different types of averaging on the convergence of the basic QuAFL pattern, on the CelebA dataset, with $n = 100$ clients. All variants execute in the same setup, with individually-tuned hyper-parameters. We clearly observe that the variant where averaging is applied both at the server and at the client performs the best, which validates our choices. In Figure 5, we compare the convergence of QuAFL with the lattice quantizer, relative to QuAFL using the standard QSGD quantizer (Alistarh et al., 2017a). We note that using QSGD is not theoretically-justified, and in fact we had to perform careful tuning in order to obtain stable convergence for this variant. It is interesting that QuAFL appears to support this, albeit at the cost of slower convergence.

Finally, in Figure 6, we compare QuAFL convergence relative to FedBuff, a state-of-the-art *asynchronous* FL protocol (Nguyen et al., 2022a), which performs buffering of client messages at the server, updating the server model as soon as $s$ messages, with $K$ local updates each, are in the server buffer. Since the timing model of FedBuff is different, we compare convergence in terms of total number of gradient steps taken by clients, without quantization. We observe that QuAFL converges faster: our analysis suggests that this is because QuAFL takes into account *partial progress by slow clients*, whereas in FedBuff slow clients constantly contribute less significantly to the server updates. We present additional experimental results in the Appendix, specifically on higher node counts (up to 300), full-convergence experiments, as well as across all other tasks.

## 6 CONCLUSIONS AND LIMITATIONS

We have provided the first variant of FedAvg which incorporates both asynchronous and compressed communication, and have shown that this algorithm can still provide good convergence guarantees. Our analysis should be extensible to more complex federated optimizers, such as gradient tracking, e.g. (Haddadpour et al., 2021), controlled averaging (Karimireddy et al., 2020), or variance-reduced variants (Gorbunov et al., 2021). Our work has the following limitations. First, our algorithm has an optimal convergence rate when $H = \Theta(K)$, which we believe is natural due to asynchrony. Second, this version of the analysis requires the expected number of local steps $H$ to be the same across all devices. We believe that this can be addressed either by modifying the objective, or by de-biasing via sampling, and plan to investigate it in future work, together with validation on real-world deployments.

# 7 REPRODUCIBILITY STATEMENT

All the code required to reproduce our experimental setup and our experiments is available at https://anonymous.4open.science/r/QuAFL-Anonymous.

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

CONTENTS

## A   EXPERIMENTAL SETUP

In this section, we describe our experimental setup in detail. We begin by defining the hyper-parameters which control the behavior of QuAFL and FedAvg. Then, we proceed by carefully describing the way in which we simulated each of the algorithms. Finally, we detail the datasets, tasks, and models used for our experiments.

### A.1   HYPER-PARAMETERS

We first define our hyper-parameters; in the later sections, we will examine their impact on algorithm behavior through ablation studies.

$n$: Number of the clients.

$s$: Number of clients interacting with the server at each step.

$K$: *In QuAFL*, this is the maximum number of allowed local steps by each client between two server calls. *In FedAvg*, this is the number of local steps performed by each client upon each server call.

$b$: Number of bits used to send a coordinate after quantization.

$swt$: Server waiting time, i.e. the amount of time that server waits between two consecutive calls.

$sit$: Server interaction time, i.e. the amount of time that server needs to send and receive necessary data (excluding computation time).

### A.2   SIMULATION

We attempt to simulate a realistic FL deployment scenario, as follows. We assume a *server* and *n clients*, each of which initially has a model copy. The training dataset is distributed among the clients so that each of them has access to $1/n$ of the training data. We track the performance of each algorithm by evaluating the server's model, on an unseen validation dataset. We measure *loss* and *accuracy* of the model with respect to *simulation time*, *server steps*, and *total local steps performed by clients*. These setups so far were common between QuAFL and FedAvg. In the following, we are going to describe their specifications and differences.

**QuAFL:** Upon each server call, the server chooses $s$ clients uniformly at random. It then sends its model to those clients and asks for their current local models. (Recall that clients send their model immediately to the server.) Each of the clients will have taken a maximum of $K$ local steps by the time it is contacted by the server. The server then replaces its model with a carefully-computed average over the received models and its current model. This process increases time on the server by the *server interaction time (sit)*. The server then waits for another interval of *server waiting time (swt)* to make its next call. The $s$ receiving clients replace their model with the weighted average between their current model and the received server's model. Since each client performs local steps from its last interaction time until the current server time, nodes are effectively executing asynchronously. Moreover, note that communication is compressed, as all the models get encoded in their source and decoded in their destination.

> **Quantization:** To have a lightweight but efficient communication between clients and the server, we use the well-known lattice quantization (Davies et al., 2021). Using this method, we send $b$ instead of 32 bits for each scalar dimension. Informally, each 32-bit number maps to one of the $2^b$ quantized levels and can be sent using $b$ bits only. The encoded number can then be decoded to a sufficiently close number at the destination, following the quantization protocol.

**FedAvg:** In the beginning of each round, server chooses $s$ clients randomly, and sends its current model to them. Each of those clients receives the model, uncompressed, and performs exactly $K$ local steps using this model as the starting point, and then sends back the resulting model to the server. The server then computes the average of the received models and adopts it as its model. By this synchronous structure, in each round, the server must wait for the *slowest* client to complete its local steps plus an extra *sit* for the communication time. After completing each round, the server starts the next call immediately, that means $swt = 0$ in FedAvg.

**Timing Experiments.** We differentiate between two types of timing experiments. *Uniform timing* experiments, presented in the paper body, assume all clients take the same amount of time for a

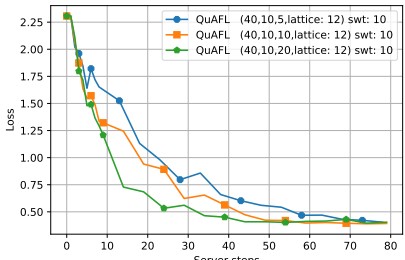

**Figure 7:** Impact of the maximum number of local steps $K \in \{5, 10, 20\}$ on the QuAFL algorithm / Fashion MNIST.

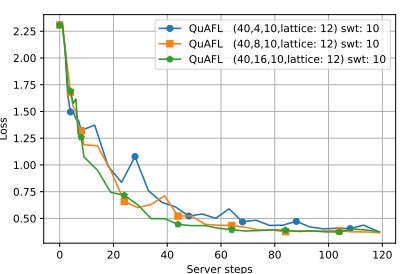

**Figure 8:** Impact of the number of interacting peers $s \in \{4, 8, 16\}$ on the convergence of the algorithm.

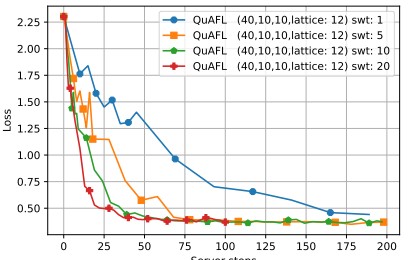

**Figure 9:** Impact of the server contact frequency (controlled via server timeout *swt*) on the convergence of the algorithm.

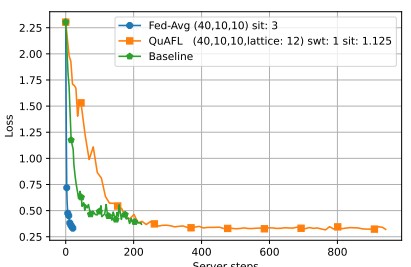

**Figure 10:** Convergence comparison relative to total number of rounds, between QuAFL, FedAvg, and the sequential baseline.

gradient step. However, in real-world setups, different devices may require different amounts of time to perform a single local step. This is one of the main disadvantages of synchronous federated optimization algorithms. To demonstrate how this fact affects the experiments, in our *Non-uniform* timing experiments we differentiate clients to be either *fast* or *slow*. The length of each local step can be characterized as a memoryless time event. Therefore, the length of each local step can be defined by a random variable $X \sim exponential(\lambda)$. The parameter $\lambda$ is $1/2$ for *fast clients* and $1/8$ for *slow* clients; the expected runtime $\mathbb{E}(X)$ would be 2 and 8, respectively. In each timing experiment, we assumed only one fourth of clients to be *slow*.

## A.3 DATASETS AND MODELS

We used Pytorch to manage the training process in our algorithm. We have trained neural networks for image classification tasks on three well-known datasets, **MNIST**, **Fashion MNIST**, and **CIFAR-10**. For all the datasets, we used the default train/test split of the dataset for our training/validation dataset. In the following, we describe the model architecture and the training hyper-parameters used to train on each of these datasets.

**MNIST:** We used SGD optimizer with constant $lr = 0.5$ in all the training process. We used a two-layer MLP architecture with (784,32,10) nodes in its layers respectively. We used batch size 128 in each client's SGD step.

**Fashion MNIST:** Although this dataset has the same sample size and number of classes as **MNIST**, obtaining competitive performance on it requires a more complicated architecture. Therefore, we used a CNN model to train on this model and demonstrated the performance of our algorithm in a non-convex task. To optimize the models, we used Adam optimizer with constant $lr = 0.001$ and batch size 100.

**CIFAR-10:** To load this dataset, we used data augmentation and normalization. For this task, we trained ResNet20 models. Moreover, the SGD optimizer with constant $lr = 0.03$ is used to in the training process. The batch size 64/200 is used for training/validation.

## A.4 RESULTS ON FASHION MNIST (FMNIST)

We begin by validating our earlier results, presented in the paper body, for the slightly more complex FMNIST dataset, and on a convolutional model.

In Figures 7 and 8 we examine the impact of the parameters $K$ and $s$, respectively, on the total number of interaction rounds at the server, to reach a certain training loss. As expected, we notice

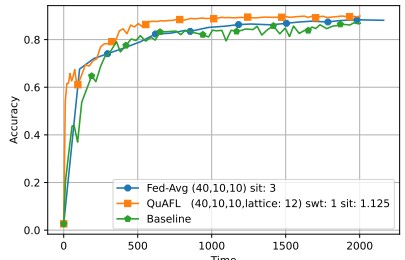

**Figure 11:** Time vs. accuracy for various algorithm variants, on Fashion MNIST.

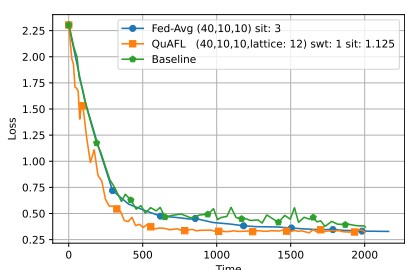

**Figure 12:** Timing vs. loss for various algorithm variants, on Fashion MNIST.

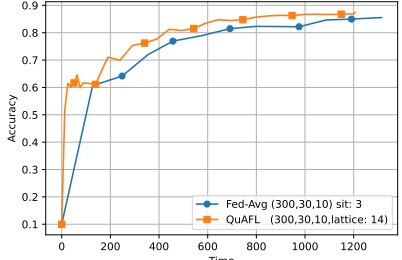

**Figure 13:** Time vs. accuracy for n=300 clients, s=30 peers on Fashion MNIST.

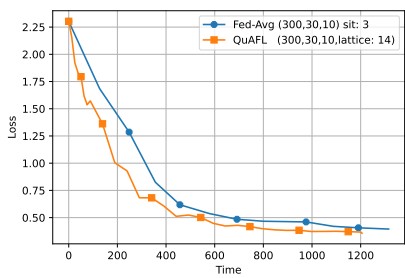

**Figure 14:** Timing vs. loss for n=300 clients, s=30 peers on Fashion MNIST.

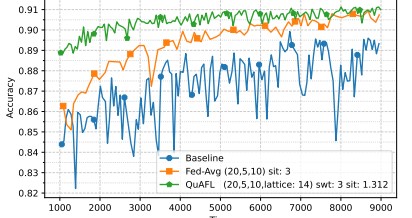

**Figure 15:** Full convergence result for $n = 20$ clients and $s = 5$ on F-MNIST. All methods eventually reach the sequential $\sim 91\%$ top-1 accuracy on this task, but QuAFL is the fastest to do so in terms of wall-clock time.

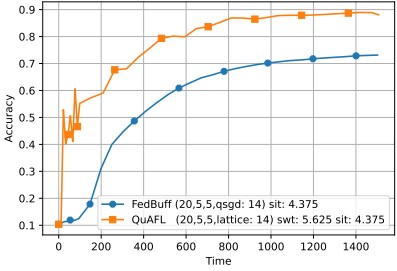

**Figure 16:** Validation accuracy for QuAFL with Lattice vs FedBuff with QSGD quantization, at the same bit width. QuAFL shows significantly better convergence.

that higher $K$ and $s$ improve the convergence behavior of the algorithm. In Figures 9 we examine the impact of the server waiting time on the convergence of the algorithm relative to the number of server rounds. Again, we notice that a higher server waiting time improves convergence, as it allows the server to take advantage of additional local steps performed at the clients, as predicted by our analysis. (Higher $swt$ means higher average number of steps completed $H$.)

Next, we examine the convergence, again in terms of number of optimization "rounds" at the server, between the sequential Baseline, FedAvg, and QuAFL. As expected, the Baseline is faster to converge than FedAvg, which in turn is faster than QuAFL in this measure. Specifically, the difference between QuAFL and the other algorithms comes because of the fact that, in our algorithm, nodes operate on *old* variants of the model at every step, which slows down convergence. Next, we examine convergence in terms of actual time, in the heterogeneous setting in which 25% of the clients are slow.

In Figure 11, we observe the validation accuracy ensured by various algorithms relative to the simulated execution time, whereas in Figure 12 we observe the training loss versus the same metric. (We assume that, in *Baseline*, a single node acts as both the client and the server, and that this node is slow, i.e. has higher per-step times.) To further support the robustness of our algorithm in regimes with large number of clients, we conducted an experiment with $n = 300$ clients and $s = 30$ peers interacting with the server at each step. The validation accuracy and loss versus time regarding the mentioned experiment plotted in Figure 13 and Figure 14 respectively. We observe that, importantly, if *time* is taken into account rather than the number of server rounds, QuAFL can provide notable

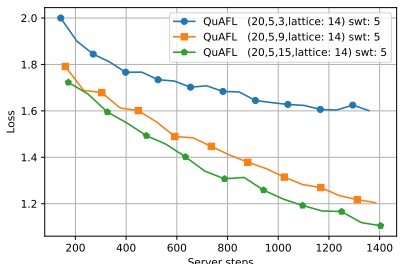

**Figure 17:** Impact of maximum local steps $K \in \{3, 9, 15\}$ on the QuAFL algorithm, on ResNet20/CIFAR-10.

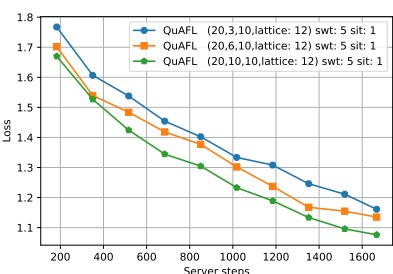

**Figure 18:** Impact of the number of interacting peers $s \in \{3, 6, 10\}$ on the convergence of the algorithm.

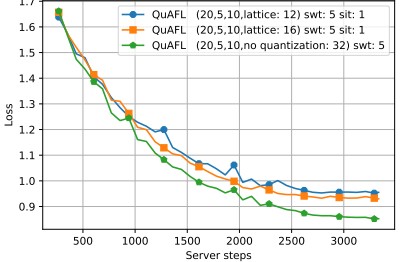

**Figure 19:** Impact of the number of bits for quantization $b \in \{12, 16, 32\}$ on the convergence of the algorithm.

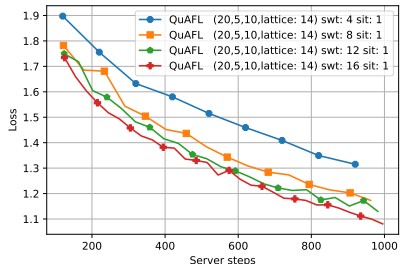

**Figure 20:** Impact of the server contact frequency (controlled via server timeout *swt*) on the convergence of the algorithm.

speedups in these metrics. This is specifically because of its asynchronous communication patters, which allow it to complete rounds faster, without having to always wait for the slow nodes to complete their local computation. While this behaviour is simulated, we believe that this reflects the algorithm's practical potential. Finally, Figure 15 shows that all methods can reach the maximum accuracy for this dataset/model combination (for the SGD baseline, this occurs later), although QuAFL is the fastest to do so in terms of wall-clock time.

Our last experiment in Figure 16 examines whether naive QSGD quantization of the transmitted updates in FedBuff (Nguyen et al., 2022a) can converge at a good rate, relative to QuAFL at the same quantization ratio. We find that this is not the case: first, we remark that, with careful tuning of the learning rate, FedBuff can indeed converge. We believe that this is because of the specific application: the norm of the model and updates in DNNs tends to be small, and therefore the quantization error induced by direct QSGD stochastic quantization is also manageable. However, we observe a clear loss of convergence for the FedBuff + QSGD algorithm in this case, relative to QuAFL, which we ascribe to the fact that we are essentially running a heuristic.

## A.5 RESULTS ON CIFAR-10

We now present results for a standard image classification task on the CIFAR-10 dataset, using a ResNet20 model (He et al., 2016).

Figures 17 and 18 show the decrease in training loss versus the number of server steps (or rounds) for different values of $K$ and $s$ respectively. As our theory suggests, increasing $K$ and $s$ leads to an improvement in the convergence rate of the system. Figure 19 demonstrates the impact of the number of quantization bits $b$, on the convergence behaviour of the algorithm. According to the definition of $b$, increasing the number of quantization bits improves the communication accuracy. Thus, as it can be seen in the graph, higher values of $b$ enhance the convergence relative to the number of server steps. Finally, Figure 20 shows the impact of the server interaction frequency, again controlled via the timeout parameter *swt*, on the algorithm's convergence. It is apparent that a very high interaction frequency can slow the algorithm down, by not allowing it to take advantage of the clients' local steps.

In Figures 21 and 22, we examine the validation accuracy and loss, respectively, ensured by various algorithms versus the simulated execution time. (As in the F-MNIST experiments, we assumed the *Baseline* to be a single slow node that performs an optimization step per round.) Again, the asynchronous nature of QuAFL provides a faster convergence rate than its synchronous counterparts; which can be clearly seen in the mentioned figures.

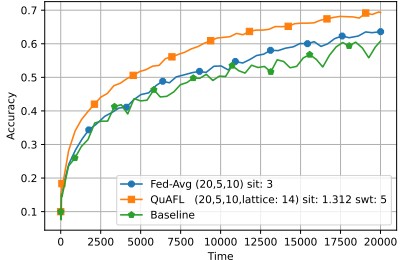
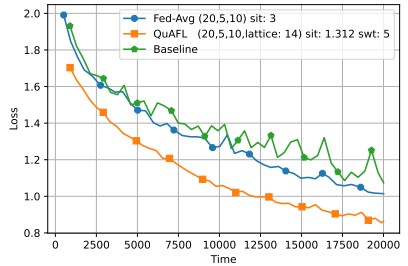

**Figure 21:** Time vs. validation accuracy for various algorithm variants.

**Figure 22:** Timing versus validation loss for various algorithm variants.

## B  THE COMPLETE ANALYSIS

### B.1  OVERVIEW AND NOTATION

Recall that $X_t$ denotes the model of the server at step $t$, and $X^i$ is the local model of client $i$ after its last interaction with the server. Also, $\widetilde{h}_i$ is the sum of local gradient steps for model $X^i$ since its last interaction with the server.

For the convergence analysis, local steps of the clients that are not selected by the server don't have any effect on the server or other clients. Therefore we do not need to assume that clients are doing their local steps asynchronous, and we can assume that all clients run their local gradient steps after the server contacts them. The only thing that we should consider is the randomness of the server selecting the clients, and the fact that the server can contact nodes before they have finished their $K$ steps. For this purpose, we assume that their number of steps is a random number $H_t^i$ with mean $H$. To show the analysis in this setting, we introduce new notations that consider the server round. To this end, we use $X_t^i$ as the value of $X^i$ when the server is running its $t$th iteration, And $\widetilde{h}_{i,t}$ for the sum of local steps at this time. We show each local step $q$ with a superscript. Formally, we have

$$\widetilde{h}_{i,t}^0 = 0.$$

and for $1 \leq q \leq H_t^i$ let:

$$\widetilde{h}_{i,t}^q = \widetilde{g}_i(X_t^i - \sum_{s=0}^{q-1} \eta \widetilde{h}_{i,t}^s),$$

and

$$\widetilde{h}_{i,t} = \sum_{q=0}^{H_t^i} \widetilde{h}_{i,t}^q$$

Further , for $1 \leq q \leq H_t^i$, let

$$h_{i,t}^q = \mathbb{E}[\widetilde{g}_i(X_t^i - \sum_{s=0}^{q-1} \eta \widetilde{h}_{i,t}^s)] = \nabla f(X_t^i - \sum_{s=0}^{q-1} \eta \widetilde{h}_{i,t}^s)$$

be the expected value of $\widetilde{h}_{i,t}^q$ taken over the randomness of the stochastic gradient $\widetilde{g}_i$. Also, we have:

$$h_{i,t} = \sum_{q=0}^{H_t^i} h_{i,t}^q$$

### B.2  PROPERTIES OF LOCAL STEPS

**Lemma B.1.** *For any agent $i$ and step $t$*

$$\mathbb{E}\|h_{i,t}^q\|^2 \leq \frac{\sigma^2}{K^2} + 8L^2\mathbb{E}\|X_t^i - \mu_t\|^2 + 4\mathbb{E}\|\nabla f_i(\mu_t)\|^2.$$

*Proof.*

$$\mathbb{E}\|h_{i,t}^q\|^2 \leq \mathbb{E}\left\|\left(\nabla f_i(X_t^i - \sum_{s=0}^{q-1} \eta \widetilde{h}_{i,t}^s) - \nabla f_i(\mu_t)\right) + \nabla f_i(\mu_t)\right\|^2$$

$$\leq 2\mathbb{E}\left\|\nabla f_i(X_t^i - \sum_{s=0}^{q-1} \eta \widetilde{h}_{i,t}^s) - \nabla f_i(\mu_t)\right\|^2 + 2\mathbb{E}\|\nabla f_i(\mu_t)\|^2$$

$$\leq 4L^2\mathbb{E}\|X_t^i - \mu_t\|^2 + 4\eta^2 L^2 q \sum_{s=0}^{q-1} \mathbb{E}\|\widetilde{h}_{i,t}^s\|^2 + 2\mathbb{E}\|\nabla f_i(\mu_t)\|^2$$

$$\leq 4L^2\mathbb{E}\|X_t^i - \mu_t\|^2 + 4\eta^2 L^2 q \sum_{s=0}^{q-1} (\mathbb{E}\|h_{i,t}^s\|^2 + \sigma^2) + 2\mathbb{E}\|\nabla f_i(\mu_t)\|^2$$

the rest of the proof is done by induction, and assuming $\eta < \frac{1}{4LK^2}$. $\qquad\square$

**Lemma 4.6.** *For any step $t$, we have that*

$$\sum_{i=1}^n \mathbb{E}\|\widetilde{h}_{i,t}\|^2 \leq 2nK(\sigma^2 + 2KG^2) + 8L^2K^2\mathbb{E}[\Phi_t] + 4nK^2B^2\mathbb{E}\|\nabla f(\mu_t)\|^2.$$

*Proof.* Using lemma B.1

$$\sum_{i=1}^n \mathbb{E}\|\widetilde{h}_{i,t}\|^2 = \sum_{i=1}^n \sum_{h=0}^K Pr[H_t^i = h]\mathbb{E}\|\sum_{q=1}^h \widetilde{h}_{i,t}^q\|^2$$

$$\leq \sum_{i=1}^n \sum_{h=1}^K Pr[H_t^i = h]h \sum_{q=1}^h \mathbb{E}\|\widetilde{h}_{i,t}^q\|^2$$

$$\leq nK\sigma^2 + \sum_{i=1}^n \sum_{h=1}^K Pr[H_t^i = h]h \sum_{q=1}^h \mathbb{E}\|h_{i,t}^q\|^2$$

$$\leq nK\sigma^2 + \sum_{i=1}^n K^2\left(\frac{\sigma^2}{K^2} + 8L^2\mathbb{E}\|X_t^i - \mu_t\|^2 + 4\mathbb{E}\|\nabla f_i(\mu_t)\|^2\right)$$

$$\leq 2nK\sigma^2 + \sum_{i=1}^n K^2\left(8L^2\mathbb{E}\|X_t^i - \mu_t\|^2 + 4\mathbb{E}\|\nabla f_i(\mu_t)\|^2\right)$$

$$\leq 2nK\sigma^2 + 8L^2K^2\mathbb{E}[\Phi_t] + 4nK^2G^2 + 4nK^2B^2\mathbb{E}\|\nabla f(\mu_t)\|^2.$$

$\qquad\square$

**Lemma B.2.** *For any local step $1 \leq q$, and agent $1 \leq i \leq n$ and step $t$*

$$\mathbb{E}\|\nabla f_i(\mu_t) - h_{i,t}^q\|^2 \leq 4L^2\eta^2 q^2\sigma^2 + 4L^2\mathbb{E}\|X_t^i - \mu_t\|^2 + 8L^2\eta^2 q^2\mathbb{E}\|\nabla f_i(\mu_t)\|^2.$$

*Proof.*

$$\mathbb{E}\|\nabla f_i(\mu_t) - h_{i,t}^q\|^2 = \mathbb{E}\|\nabla f_i(\mu_t) - \nabla f_i(X_t^i - \sum_{s=0}^{q-1} \eta \widetilde{h}_{i,t}^s)\|^2$$

$$\leq L^2\mathbb{E}\|\mu_t - X_t^i + \sum_{s=0}^{q-1} \eta \widetilde{h}_{i,t}^s\|^2$$

$$\leq 2L^2\mathbb{E}\|X_t^i - \mu_t\|^2 + 2L^2\eta^2\mathbb{E}\|\sum_{s=0}^{q-1} \widetilde{h}_{i,t}^s\|^2$$

$$\leq 2L^2\mathbb{E}\|X_t^i - \mu_t\|^2 + 2L^2\eta^2 q \sum_{s=0}^{q-1} \mathbb{E}\|\widetilde{h}_{i,t}^s\|^2$$

$$\overset{\text{Lemma (B.1)}}{\leq} 2L^2\mathbb{E}\|X_t^i - \mu_t\|^2 + 2L^2\eta^2 q^2\left(2\sigma^2 + 8L^2\mathbb{E}\|X_t^i - \mu_t\|^2\right.$$

$$\left. + 4\mathbb{E}\|\nabla f_i(\mu_t)\|^2\right)$$

$$= 4L^2\eta^2 q^2\sigma^2 + (2L^2 + 16L^4\eta^2 q^2)\mathbb{E}\|X_t^i - \mu_t\|^2$$
$$+ 8L^2\eta^2 q^2\mathbb{E}\|\nabla f_i(\mu_t)\|^2$$
$$\le 4L^2\eta^2 q^2\sigma^2 + 4L^2\mathbb{E}\|X_t^i - \mu_t\|^2 + 8L^2\eta^2 q^2\mathbb{E}\|\nabla f_i(\mu_t)\|^2$$

and the last inequality comes from $\eta < \frac{1}{4LK}$. □

**Lemma B.3.** *For any time step $t$*

$$\sum_{i=1}^{n}\mathbb{E}\langle\nabla f(\mu_t), -h_{i,t}\rangle \le 4KL^2\mathbb{E}[\Phi_t] + (-\frac{3Hn}{4} + 8B^2L^2\eta^2 K^3 n)\mathbb{E}\|\nabla f(\mu_t)\|^2$$
$$+ 4nL^2\eta^2 K^3(\sigma^2 + 2G^2).$$

*Proof.*

$$\sum_{i=1}^{n}\mathbb{E}\langle\nabla f(\mu_t), -h_{i,t}\rangle = \sum_{i=1}^{n}\sum_{h=1}^{K}Pr[H_t^i = h]\mathbb{E}\langle\nabla f(\mu_t), -\sum_{q=1}^{h}h_{i,t}^q\rangle + \sum_{i=1}^{n}Pr[H_t^i = 0]\mathbb{E}\langle\nabla f(\mu_t), 0\rangle$$

$$= \sum_{i=1}^{n}\sum_{h=1}^{K}Pr[H_t^i = h]\sum_{q=1}^{h}\Big(\mathbb{E}\langle\nabla f(\mu_t), \nabla f_i(\mu_t) - h_{i,t}^q\rangle - \mathbb{E}\langle\nabla f(\mu_t), \nabla f_i(\mu_t)\rangle\Big)$$

Using Young's inequality we can upper bound $\mathbb{E}\langle\nabla f(\mu_t), \nabla f_i(\mu_t) - h_{i,t}^q\rangle$ by

$$\frac{\mathbb{E}\|\nabla f(\mu_t)\|^2}{4} + \mathbb{E}\Big\|\nabla f_i(\mu_t) - h_{i,t}^q\Big\|^2.$$

Plugging this in the above inequality we get:

$$\sum_{i=1}^{n}\mathbb{E}\langle\nabla f(\mu_t), -h_{i,t}\rangle$$

$$\le \sum_{i=1}^{n}\sum_{h=1}^{K}Pr[H_t^i = h]\sum_{q=1}^{h}\Big(\mathbb{E}\|\nabla f(\mu_t) - h_{i,t}^q\|^2 + \frac{\mathbb{E}\|\nabla f(\mu_t)\|^2}{4} - \mathbb{E}\langle\nabla f(\mu_t), \nabla f_i(\mu_t)\rangle\Big)$$

$$\overset{\text{Lemma B.2}}{\le} \sum_{i=1}^{n}\sum_{h=1}^{K}Pr[H_t^i = h]\sum_{q=1}^{h}\Big(4L^2\eta^2 q^2\sigma^2 + 4L^2\mathbb{E}\|X_t^i - \mu_t\|^2 + 8L^2\eta^2 q^2\mathbb{E}\|\nabla f_i(\mu_t)\|^2$$

$$+ \frac{\mathbb{E}\|\nabla f(\mu_t)\|^2}{4} - \mathbb{E}\langle\nabla f(\mu_t), \nabla f_i(\mu_t)\rangle\Big)$$

$$\le \sum_{i=1}^{n}\sum_{h=1}^{K}Pr[H_t^i = h]h\Big(4L^2\eta^2 h^2\sigma^2 + 4L^2\mathbb{E}\|X_t^i - \mu_t\|^2 + 8L^2\eta^2 h^2\mathbb{E}\|\nabla f_i(\mu_t)\|^2$$

$$+ \frac{\mathbb{E}\|\nabla f(\mu_t)\|^2}{4} - \mathbb{E}\langle\nabla f(\mu_t), \nabla f_i(\mu_t)\rangle\Big)$$

$$\le 4KL^2\mathbb{E}[\Phi_t] + 4nL^2\eta^2 K^3(\sigma^2 + 2G^2) + (8B^2nL^2\eta^2 K^3 + \frac{Hn}{4} - Hn)\mathbb{E}\|\nabla f(\mu_t)\|^2\Big)$$

Where in the last step we used that $\mathbb{E}[H_t^i] = H$, and $\sum_{i=1}^{n}\frac{f_i(x)}{n} = f(x)$, for any vector $x$. □

## B.3 UPPER BOUNDING POTENTIAL FUNCTIONS

We proceed by proving the lemma 4.4 which upper bounds the expected change in potential:

**Lemma 4.4.** *For any time step $t$ we have:*

$$\mathbb{E}[\Phi_{t+1}] \le \left(1 - \frac{1}{4n}\right)\mathbb{E}[\Phi_t] + 8s\eta^2\sum_{i=1}^{n}\mathbb{E}\|\widetilde{h}_{i,t}\|^2 + 16n(R^2 + 7)^2\gamma^2.$$

*Proof.* First we bound change in potential $\Delta_t = \Phi_{t+1} - \Phi_t$ for some fixed time step $t > 0$.

For this, let $\Delta_t^S$ be the change in potential when set $S$ of agents wake up. for $i \in S$ define $S_t^i$ and $S_t$ as follows:

$$S_t^i = -\frac{s}{s+1}\eta\widetilde{h}_{i,t} + \frac{Q(X_t) - X_t}{s+1}$$

$$S_t = -\frac{1}{s+1}\eta\sum_{i \in S}\widetilde{h}_{i,t} + \frac{1}{s+1}\sum_{i \in S}(Q(X_t^i - \eta\widetilde{h}_{i,t}) - (X_t^i - \eta\widetilde{h}_{i,t}))$$

We have that:

$$X_{t+1}^i = \frac{sX_t^i + X_t}{s+1} + S_t^i$$

$$X_{t+1} = \frac{\sum_{i \in S}X_t^i + X_t}{s+1} + S_t$$

$$\mu_{t+1} = \mu_t + \frac{\sum_{j \in S}S_t^j + S_t}{n+1}$$

This gives us that for $i \in S$:

$$X_{t+1}^i - \mu_{t+1} = \frac{sX_t^i + X_t}{s+1} + S_t^i - \frac{\sum_{j \in S}S_t^j + S_t}{n+1} - \mu_t$$

$$X_{t+1} - \mu_{t+1} = \frac{\sum_{i \in S}X_t^i + X_t}{s+1} + S_t - \frac{\sum_{j \in S}S_t^j + S_t}{n+1} - \mu_t$$

For $k \notin S$ we get that

$$X_{t+1}^k - \mu_{t+1} = X_t^k - \frac{\sum_{j \in S}S_t^j + S_t}{n+1} - \mu_t.$$

Hence:

$$\Delta_t^S = \sum_{i \in S}\left(\left\|\frac{sX_t^i + X_t}{s+1} + S_t^i - \frac{\sum_{j \in S}S_t^j + S_t}{n+1} - \mu_t\right\|^2 - \left\|X_t^i - \mu_t\right\|^2\right)$$

$$+ \left\|\frac{\sum_{i \in S}X_t^i + X_t}{s+1} + S_t - \frac{\sum_{j \in S}S_t^j + S_t}{n+1} - \mu_t\right\|^2 - \left\|X_t - \mu_t\right\|^2$$

$$+ \sum_{k \notin S}\left(\left\|X_t^k - \frac{\sum_{j \in S}S_t^j + S_t}{n+1} - \mu_t\right\|^2 - \left\|X_t^k - \mu_t\right\|^2\right)$$

$$= \sum_{i \in S}\left(\left\|\frac{sX_t^i + X_t}{s+1} - \mu_t\right\|^2 + \left\|S_t^i + \frac{\sum_{j \in S}S_t^j + S_t}{n+1}\right\|^2\right.$$

$$\left. + 2\left\langle\frac{sX_t^i + X_t}{s+1} - \mu_t, S_t^i - \frac{\sum_{j \in S}S_t^j + S_t}{n+1}\right\rangle - \left\|X_t^i - \mu_t\right\|^2\right)$$

$$+ \left(\left\|\frac{\sum_{i \in S}X_t^i + X_t}{s+1} - \mu_t\right\|^2 + \left\|S_t - \frac{\sum_{j \in S}S_t^j + S_t}{n+1}\right\|^2\right.$$

$$\left. + 2\left\langle\frac{\sum_{i \in S}X_t^i + X_t}{s+1} - \mu_t, S_t - \frac{\sum_{j \in S}S_t^j + S_t}{n+1}\right\rangle - \left\|X_t - \mu_t\right\|^2\right)$$

$$+ \sum_{k \notin S}2\left\langle X_t^k - \mu_t, -\frac{\sum_{j \in S}S_t^j + S_t}{n+1}\right\rangle + \sum_{k \notin S}\left\|\frac{\sum_{j \in S}S_t^j + S_t}{n+1}\right\|^2$$

Observe that:

$$\sum_{k=0}^{n}\left\langle X_t^k - \mu_t, -\frac{\sum_{j \in S}S_t^j + S_t}{n+1}\right\rangle = 0.$$

After combining the above two equations, we get that:

$$\Delta_t^S = \sum_{i\in S}\left(\left\|\frac{s(X_t^i-\mu_t)+(X_t-\mu_t)}{s+1}\right\|^2 - \frac{s}{s+1}\left\|X_t^i-\mu_t\right\|^2 - \frac{1}{s+1}\left\|X_t-\mu_t\right\|^2\right)$$

$$+\left(\left\|\frac{\sum_{i\in S}(X_t^i-\mu_t)+(X_t-\mu_t)}{s+1}\right\|^2 - \sum_{i\in S}\frac{1}{s+1}\left\|X_t^i-\mu_t\right\|^2 - \frac{1}{s+1}\left\|X_t-\mu_t\right\|^2\right)$$

$$+\sum_{i\in S}\left(\left\|S_t^i-\frac{\sum_{j\in S}S_t^j+S_t}{n+1}\right\|^2 + 2\left\langle\frac{sX_t^i+X_t}{s+1}-\mu_t,S_t^i\right\rangle\right)$$

$$+\left\|S_t-\frac{\sum_{j\in S}S_t^j+S_t}{n+1}\right\|^2 + 2\left\langle\frac{\sum_{i\in S}X_t^i+X_t}{s+1}-\mu_t,S_t\right\rangle$$

$$+\sum_{k\notin S}\left\|\frac{\sum_{j\in S}S_t^j+S_t}{n+1}\right\|^2$$

By simplifying the above, we get:

$$\Delta_t^S = \frac{-s}{(s+1)^2}\sum_{i\in S}\|X_t^i-X_t\|^2 - \frac{1}{(s+1)^2}\sum_{i\in S}\|X_t^i-X_t\|^2 - \frac{1}{(s+1)^2}\sum_{i,j\in S}\|X_t^i-X_t^j\|^2)$$

$$+\sum_{i\in S}\left\|S_t^i-\frac{\sum_{j\in S}S_t^j+S_t}{n+1}\right\|^2 + \frac{2s}{s+1}\sum_{i\in S}\left\langle X_t^i-\mu_t,S_t^i\right\rangle + \frac{2}{s+1}\sum_{i\in S}\left\langle X_t-\mu_t,S_t^i\right\rangle$$

$$+\left\|S_t-\frac{\sum_{j\in S}S_t^j+S_t}{n+1}\right\|^2 + \frac{2}{s+1}\sum_{i\in S}\left\langle X_t^i-\mu_t,S_t\right\rangle + \frac{2}{s+1}\left\langle X_t-\mu_t,S_t\right\rangle$$

$$+\sum_{k\notin S}\left\|\frac{\sum_{j\in S}S_t^j+S_t}{n+1}\right\|^2$$

Let $\alpha$ be a parameter we will fix later:

$$\left\langle X_t^i-\mu_t,S_t^i\right\rangle \overset{\text{Young}}{\leq} \alpha\left\|X_t^i-\mu_t\right\|^2 + \frac{\left\|S_t^i\right\|^2}{4\alpha}$$

Finally, we get that

$$\Delta_t^S \leq \frac{-1}{s+1}\sum_{i\in S}\|X_t^i-X_t\|^2 + 2\sum_{i\in S}\left\|S_t^i\right\|^2 + \frac{2s(s+1)}{(n+1)^2}\sum_{j\in S}\left\|S_t^j\right\|^2 + \frac{2s(s+1)}{(n+1)^2}\left\|S_t\right\|^2$$

$$+\sum_{i\in S}\frac{2s\alpha}{s+1}\left\|X_t^i-\mu_t\right\|^2 + \sum_{i\in S}\frac{s\left\|S_t^i\right\|^2}{2\alpha(s+1)} + \sum_{i\in S}\frac{2\alpha}{s+1}\left\|X_t-\mu_t\right\|^2 + \sum_{i\in S}\frac{\left\|S_t^i\right\|^2}{2\alpha(s+1)}$$

$$+2\left\|S_t\right\|^2 + \frac{2(s+1)}{(n+1)^2}\sum_{j\in S}\left\|S_t^j\right\|^2 + \frac{2(s+1)}{(n+1)^2}\left\|S_t\right\|^2 + \sum_{i\in S}\frac{2\alpha}{s+1}\|X_t^i-\mu_t\|^2 + \sum_{i\in S}\frac{\left\|S_t\right\|^2}{2\alpha(s+1)}$$

$$+\frac{2\alpha}{s+1}\|X_t-\mu_t\|^2 + \frac{\left\|S_t\right\|^2}{2\alpha(s+1)} + \sum_{j\in S}\frac{(n-s)(s+1)}{(n+1)^2}\left\|S_t^j\right\|^2 + \frac{(n-s)(s+1)}{(n+1)^2}\left\|S_t\right\|^2$$

$$=\frac{-1}{s+1}\sum_{i\in S}\|X_t^i-X_t\|^2 + (2+\frac{2(s+1)^2}{(n+1)^2}+\frac{1}{2\alpha}+\frac{(n-s)(s+1)}{(n+1)^2})\sum_{j\in S}\left\|S_t^j\right\|^2 +$$

$$(2+\frac{2(s+1)^2}{(n+1)^2}+\frac{1}{2\alpha}+\frac{(n-s)(s+1)}{(n+1)^2})\left\|S_t\right\|^2 + \sum_{i\in S}2\alpha\left\|X_t^i-\mu_t\right\|^2 + 2\alpha\left\|X_t-\mu_t\right\|^2$$

$$\leq \frac{-1}{s+1}\sum_{i\in S}\|X_t^i - X_t\|^2 + (4 + \frac{1}{2\alpha})\sum_{i\in S}\left\|S_t^i\right\|^2 +$$

$$(4 + \frac{1}{2\alpha})\left\|S_t\right\|^2 + \sum_{i\in S}2\alpha\left\|X_t^i - \mu_t\right\|^2 + 2\alpha\left\|X_t - \mu_t\right\|^2$$

Using definitions of $S_t^i$ and $S_t$, Cauchy-Schwarz inequality and properties of quantization we get that

$$\|S_t^i\|^2 \leq \frac{2s^2}{(s+1)^2}\eta^2\|\widetilde{h}_{i,t}\|^2 + \frac{2(R^2+7)^2\gamma^2}{(s+1)^2}.$$

$$\|S_t\|^2 \leq \frac{2s}{(s+1)^2}\eta^2\sum_{i\in S}\|\widetilde{h}_{i,t}\|^2 + \frac{2s^2(R^2+7)^2\gamma^2}{(s+1)^2}$$

Next, we plug this in the previous inequality:

$$\Delta_t^S \leq \frac{-1}{s+1}\sum_{i\in S}\|X_t^i - X_t\|^2 + \sum_{i\in S}2\alpha\left\|X_t^i - \mu_t\right\|^2 + 2\alpha\left\|X_t - \mu_t\right\|^2$$

$$+ (4 + \frac{1}{2\alpha})\frac{2s^2 + 2s}{(s+1)^2}\eta^2\|\widetilde{h}_{i,t}\|^2 + \frac{(2s^2+2s)(R^2+7)^2\gamma^2}{(s+1)^2}$$

$$\leq \frac{-1}{s+1}\sum_{i\in S}\|X_t^i - X_t\|^2 + \sum_{i\in S}2\alpha\left\|X_t^i - \mu_t\right\|^2 + 2\alpha\left\|X_t - \mu_t\right\|^2$$

$$+ (4 + \frac{1}{2\alpha})(\eta^2\sum_{i\in S}\|\widetilde{h}_{i,t}\|^2 + 2(R^2+7)^2\gamma^2)$$

Next, we calculate probability of choosing the set $S$ and upper bound $\Delta_t$ in expectation, for this we define $\mathbb{E}_t$ as expectation conditioned on the entire history up to and including step $t$

$$\mathbb{E}_t[\Delta_t] = \sum_S \frac{1}{\binom{n}{s}}\mathbb{E}_t[\Delta_t^S]$$

$$\leq \sum_S \frac{1}{\binom{n}{s}}\left(\frac{-1}{s+1}\sum_{i\in S}\|X_t^i - X_t\|^2 + \sum_{i\in S}2\alpha\left\|X_t^i - \mu_t\right\|^2 + 2\alpha\left\|X_t - \mu_t\right\|^2\right.$$

$$\left. + (4 + \frac{1}{2\alpha})(\eta^2\sum_{i\in S}\|\widetilde{h}_{i,t}\|^2 + 2(R^2+7)^2\gamma^2)\right)$$

$$= \frac{-\binom{n-1}{s-1}}{(s+1)\binom{n}{s}}\sum_i\|X_t^i - X_t\|^2 + \sum_i\frac{2\alpha\binom{n-1}{s-1}}{\binom{n}{s}}\left\|X_t^i - \mu_t\right\|^2 + 2\alpha\left\|X_t - \mu_t\right\|^2$$

$$+ (4 + \frac{1}{2\alpha})(\eta^2\frac{\binom{n-1}{s-1}}{\binom{n}{s}}\sum_i\|\widetilde{h}_{i,t}\|^2 + 2(R^2+7)^2\gamma^2)$$

$$\leq -\sum_i\frac{s\|X_t^i - \mu_t\|^2}{(s+1)n} + \sum_i 2\frac{s\alpha}{n}\left\|X_t^i - \mu_t\right\|^2 + 2\alpha\left\|X_t - \mu_t\right\|^2$$

$$+ (8 + \frac{1}{\alpha})(R^2+7)^2\gamma^2 + \sum_i\frac{s}{n}(4 + \frac{1}{2\alpha})\eta^2\mathbb{E}_t\|\widetilde{h}_{i,t}\|^2$$

$$\leq (\frac{-s}{(s+1)n} + 2\alpha)\Phi_t + (8 + \frac{1}{\alpha})(R^2+7)^2\gamma^2 + \sum_i\frac{s}{n}(4 + \frac{1}{2\alpha})\eta^2\mathbb{E}_t\|\widetilde{h}_{i,t}\|^2$$

By setting $\alpha = \frac{3s-1}{n(8s+8)} \geq \frac{1}{8n}$, we get that:

$$\mathbb{E}_t[\Delta_t] \leq -\frac{1}{4n}\Phi_t + 16n(R^2+7)^2\gamma^2 + \sum_i 8s\eta^2\mathbb{E}_t\|\widetilde{h}_{i,t}\|^2.$$

Next we remove the conditioning , and use the definitions of $\Delta_i$ and $S_t^i$ (for $S_t^i$ we also use upper bound which come from the properties of quantization).

$$\mathbb{E}[\mathbb{E}_t[\Phi_{t+1}]] = \mathbb{E}[\Delta_t + \Phi_t] \leq (1 - \frac{1}{4n})\mathbb{E}[\Phi_t] + 16n(R^2 + 7)^2\gamma^2 + 8s\eta^2 \sum_i \mathbb{E}\|\widetilde{h}_{i,t}\|^2$$

$\square$

**Lemma B.4.** *For any time step $t$ we have:*

$$\mathbb{E}[\Phi_{t+1}] \leq (1 - \frac{1}{5n})\mathbb{E}[\Phi_t] + 16n(R^2 + 7)^2\gamma^2 + 16nsK\eta^2(\sigma^2 + 2KG^2) + 32B^2nsK^2\eta^2\mathbb{E}\|\nabla f(\mu_t)\|^2$$

*Proof.* By combining Lemma 4.4 and 4.6 we have:

$$\begin{aligned}
\mathbb{E}[\Phi_{t+1}] &\leq (1 - \frac{1}{4n})\mathbb{E}[\Phi_t] + 16n(R^2 + 7)^2\gamma^2 + \\
&\qquad 8s\eta^2\Big(2nK(\sigma^2 + 2KG^2) + 8L^2K^2\mathbb{E}[\Phi_t] + 4nK^2B^2\mathbb{E}\|\nabla f(\mu_t)\|^2\Big) \\
&= (1 - \frac{1}{4n} + 64sL^2K^2\eta^2)\mathbb{E}[\Phi_t] + 16n(R^2 + 7)^2\gamma^2 + \\
&\qquad 16nsK\eta^2(\sigma^2 + 2KG^2) + 32B^2nsK^2\eta^2\mathbb{E}\|\nabla f(\mu_t)\|^2 \\
&\leq (1 - \frac{1}{5n})\mathbb{E}[\Phi_t] + 16n(R^2 + 7)^2\gamma^2 + \\
&\qquad 16nsK\eta^2(\sigma^2 + 2KG^2) + 32B^2nsK^2\eta^2\mathbb{E}\|\nabla f(\mu_t)\|^2
\end{aligned}$$

$\square$

**Lemma B.5.** *For the sum of potential functions in all $T$ steps we have:*

$$\sum_{t=0}^{T}\mathbb{E}[\Phi_t] \leq 80Tn^2(R^2 + 7)^2\gamma^2 + 80Tn^2sK\eta^2(\sigma^2 + 2KG^2) + 160B^2n^2sK^2\eta^2\sum_{t=0}^{T-1}\mathbb{E}\|\nabla f(\mu_t)\|^2$$

*Proof.*

$$\begin{aligned}
\sum_{t=0}^{T-1}\mathbb{E}[\Phi_{t+1}] &\leq \sum_{t=0}^{T-1}\Big((1 - \frac{1}{5n})\mathbb{E}[\Phi_t] + 16n(R^2 + 7)^2\gamma^2 + \\
&\qquad 16nsK\eta^2(\sigma^2 + 2KG^2) + 32B^2nsK^2\eta^2\mathbb{E}\|\nabla f(\mu_t)\|^2\Big) \\
&\leq (1 - \frac{1}{5n})\sum_{t=0}^{T-1}\mathbb{E}[\Phi_t] + 16Tn(R^2 + 7)^2\gamma^2 + 16TnsK\eta^2(\sigma^2 + 2KG^2) \\
&\qquad + 32B^2nsK^2\eta^2\sum_{t=0}^{T-1}\mathbb{E}\|\nabla f(\mu_t)\|^2 \\
\sum_{t=0}^{T}\mathbb{E}[\Phi_t] &\leq 5n\big(16Tn(R^2 + 7)^2\gamma^2 + 16TnsK\eta^2(\sigma^2 + 2KG^2) \\
&\qquad + 32B^2nsK^2\eta^2\sum_{t=0}^{T-1}\mathbb{E}\|\nabla f(\mu_t)\|^2\big) \\
&= 80Tn^2(R^2 + 7)^2\gamma^2 + 80Tn^2sK\eta^2(\sigma^2 + 2KG^2) \\
&\qquad + 160B^2n^2sK^2\eta^2\sum_{t=0}^{T-1}\mathbb{E}\|\nabla f(\mu_t)\|^2
\end{aligned}$$

$\square$

**Lemma 4.5.** *For any step $t$*

$$\mathbb{E}\|\mu_{t+1} - \mu_t\|^2 \leq \frac{2s^2\eta^2}{n(n+1)^2}\sum_i\mathbb{E}\left\|\widetilde{h}_{i,t}\right\|^2 + \frac{2}{(n+1)^2}(R^2 + 7)^2\gamma^2.$$

*Proof.*

$$\mathbb{E}\|\mu_{t+1} - \mu_t\|^2 \leq \sum_S \frac{1}{\binom{n}{s}(n+1)^2}\left(\mathbb{E}\Big\| - \eta \sum_{i \in S} \widetilde{h}_{i,t} + \frac{Q(X_t) - X_t}{s+1}\right.$$

$$\left. + \frac{1}{s+1}\sum_{i \in S}(Q(X_t^i - \eta\widetilde{h}_{i,t}) - (X_t^i - \eta\widetilde{h}_{i,t}))\Big\|^2\right)$$

$$\leq \sum_S \frac{1}{\binom{n}{s}(n+1)^2}\left(2s\eta^2\sum_{i \in S}\mathbb{E}\big\|\widetilde{h}_{i,t}\big\|^2 + \frac{2}{s+1}\mathbb{E}\big\|Q(X_t) - X_t\big\|^2\right.$$

$$\left. + \frac{2}{s+1}\sum_{i \in S}\mathbb{E}\big\|(Q(X_t^i - \eta\widetilde{h}_{i,t}) - (X_t^i - \eta\widetilde{h}_{i,t}))\big\|^2\right)$$

$$\leq \sum_S \frac{1}{\binom{n}{s}(n+1)^2}\left(2s\eta^2\sum_{i \in S}\mathbb{E}\big\|\widetilde{h}_{i,t}\big\|^2 + 2(R^2+7)^2\gamma^2\right)$$

$$= \sum_i \frac{2s\eta^2\binom{n-1}{s-1}}{\binom{n}{s}(n+1)^2}\mathbb{E}\big\|\widetilde{h}_{i,t}\big\|^2 + \frac{2}{(n+1)^2}(R^2+7)^2\gamma^2$$

$$= \sum_i \frac{2s^2\eta^2}{n(n+1)^2}\mathbb{E}\big\|\widetilde{h}_{i,t}\big\|^2 + \frac{2}{(n+1)^2}(R^2+7)^2\gamma^2$$

□

By plugging Lemma 4.6 in the above upper bound we get that:

**Lemma B.6.** *For any step $t$*

$$\mathbb{E}\|\mu_{t+1} - \mu_t\|^2 \leq \frac{4s^2K\eta^2(\sigma^2 + 2KG^2)}{(n+1)^2} + \frac{16s^2L^2K^2\eta^2\mathbb{E}[\Phi_t]}{n(n+1)^2}$$
$$+ \frac{8B^2s^2K^2\eta^2\mathbb{E}\|\nabla f(\mu_t)\|^2}{(n+1)^2} + \frac{2(R^2+7)^2\gamma^2}{(n+1)^2}.$$

*Proof.*

$$\mathbb{E}\|\mu_{t+1} - \mu_t\|^2 \leq \sum_i \frac{2s^2\eta^2}{n(n+1)^2}\mathbb{E}\big\|\widetilde{h}_{i,t}\big\|^2 + \frac{2}{(n+1)^2}(R^2+7)^2\gamma^2$$

$$\leq \sum_i \frac{2s^2\eta^2}{n(n+1)^2}\left(2nK(\sigma^2 + 2KG^2) + 8L^2K^2\mathbb{E}[\Phi_t] + 4nK^2B^2\mathbb{E}\|\nabla f(\mu_t)\|^2\right)$$

$$+ \frac{2(R^2+7)^2\gamma^2}{(n+1)^2}$$

$$= \frac{4s^2K\eta^2}{(n+1)^2}(\sigma^2 + 2KG^2) + \frac{16s^2L^2K^2\eta^2}{n(n+1)^2}\mathbb{E}[\Phi_t]$$

$$+ \frac{8B^2s^2K^2\eta^2}{(n+1)^2}\mathbb{E}\|\nabla f(\mu_t)\|^2 + \frac{2(R^2+7)^2\gamma^2}{(n+1)^2}$$

□

## B.4 CONVERGENCE

**Theorem B.7.** *For learning rate $\eta = \frac{n+1}{sH\sqrt{T}}$, Algorithm 1 converges at rate:*

$$\frac{1}{T}\sum_{t=0}^{T-1}\mathbb{E}\|\nabla f(\mu_t)\|^2 \leq \frac{2(f(\mu_0) - f_*)}{\sqrt{T}} + \frac{800nKL^2(R^2+7)^2\gamma^2}{H} + \frac{6KL(\sigma^2 + 2KG^2)}{H^2\sqrt{T}}$$

$$+ \frac{808n(n+1)^2K^2L^2}{sH^3T}(\sigma^2 + 2KG^2) + \frac{2(R^2+7)^2\gamma^2L\sqrt{T}}{(n+1)^2sH}$$

*Proof.* Let $\mathbb{E}_t$ denote expectation conditioned on the entire history up to and including step $t$. By $L$-smoothness we have that

$$\mathbb{E}_t[f(\mu_{t+1})] \leq f(\mu_t) + \mathbb{E}_t\langle \nabla f(\mu_t), \mu_{t+1} - \mu_t\rangle + \frac{L}{2}\mathbb{E}_t\|\mu_{t+1} - \mu_t\|^2. \tag{5}$$

First we look at $\mathbb{E}_t\langle \nabla f(\mu_t), \mu_{t+1} - \mu_t\rangle = \langle \nabla f(\mu_t), \mathbb{E}_t[\mu_{t+1} - \mu_t]\rangle$. If set $S$ is chosen at step $t + 1$, We have that

$$\mu_{t+1} - \mu_t = \frac{1}{n+1}\left(-\eta \sum_{i \in S} \widetilde{h}_{i,t} + \frac{Q(X_t) - X_t}{s+1} + \frac{1}{s+1}\sum_{i \in S}(Q(X_t^i - \eta\widetilde{h}_{i,t}) - X_t^i - \eta\widetilde{h}_{i,t}))\right)$$

Thus, in this case:

$$\mathbb{E}_t[\mu_{t+1} - \mu_t] = -\frac{\eta}{n+1}\sum_{i \in S} h_{i,t}.$$

Where we used unbiasedness of quantization and stochastic gradients. We would like to note that even though we do condition on the entire history up to and including step $t$ and this includes conditioning on $X_t^i$, the algorithm has not yet used $\widetilde{h}_{i,t}$ (it does not count towards computation of $\mu_t$), thus we can safely use all properties of stochastic gradients. Hence, we can proceed by taking into the account that each set of agents $S$ is chosen as initiator with probability $\frac{1}{\binom{n}{s}}$:

$$\mathbb{E}_t[\mu_{t+1} - \mu_t] = \sum_S \frac{1}{\binom{n}{s}}\sum_{i \in S} -\frac{\eta}{n+1}\sum_{i \in S} h_{i,t} = -\frac{s\eta}{n(n+1)}\sum_{i=1}^n h_{i,t}.$$

and subsequently

$$\mathbb{E}_t\langle \nabla f(\mu_t), \mu_{t+1} - \mu_t\rangle = \sum_{i=1}^n \frac{s\eta}{n(n+1)}\mathbb{E}_t\langle \nabla f(\mu_t), -h_{i,t}\rangle.$$

Hence, we can rewrite (5) as:

$$\mathbb{E}_t[f(\mu_{t+1})] \leq f(\mu_t) + \sum_{i=1}^n \frac{s\eta}{n(n+1)}\mathbb{E}_t\langle \nabla f(\mu_t), -h_{i,t}\rangle + \frac{L}{2}\mathbb{E}_t\|\mu_{t+1} - \mu_t\|^2.$$

Next, we remove the conditioning

$$\mathbb{E}[(\mu_{t+1})] = \mathbb{E}[\mathbb{E}_t[f(\mu_{t+1})]] \leq \mathbb{E}[f(\mu_t)] + \sum_{i=1}^n \frac{s\eta}{n(n+1)}\mathbb{E}\langle \nabla f(\mu_t), -h_{i,t}\rangle$$
$$+ \frac{L}{2}\mathbb{E}\|\mu_{t+1} - \mu_t\|^2.$$

This allows us to use Lemmas B.6 and B.3:

$$\mathbb{E}[f(\mu_{t+1})] - \mathbb{E}[f(\mu_t)] \leq \frac{s\eta}{n(n+1)}\left(4KL^2\mathbb{E}[\Phi_t] + (-\frac{3Hn}{4} + 8B^2L^2\eta^2K^3n)\mathbb{E}\|\nabla f(\mu_t)\|^2\right.$$
$$\left. + 4nL^2\eta^2K^3(\sigma^2 + 2G^2)\right)$$
$$+ \frac{L}{2}\left(\frac{4s^2K\eta^2(\sigma^2 + 2KG^2)}{(n+1)^2} + \frac{16s^2L^2K^2\eta^2\mathbb{E}[\Phi_t]}{n(n+1)^2}\right.$$
$$\left. + \frac{8B^2s^2K^2\eta^2\mathbb{E}\|\nabla f(\mu_t)\|^2}{(n+1)^2} + \frac{2(R^2 + 7)^2\gamma^2}{(n+1)^2}\right)$$
$$= \left(\frac{4\eta sKL^2}{n(n+1)} + \frac{8s^2K^2L^3\eta^2}{n(n+1)^2}\right)\mathbb{E}[\Phi_t]$$
$$+ \left(\frac{4sL^2\eta^3K^3}{n+1} + \frac{2s^2K\eta^2L}{(n+1)^2}\right)(\sigma^2 + 2KG^2) + \frac{(R^2 + 7)^2\gamma^2L}{(n+1)^2}$$
$$+ \left(\frac{-3\eta sH}{4(n+1)} + \frac{8B^2L^2\eta^3sK^3}{n+1} + \frac{4B^2s^2K^2L\eta^2}{(n+1)^2}\right)\mathbb{E}\|\nabla f(\mu_t)\|^2$$

By simplifying the above inequality we get:

$$\mathbb{E}[f(\mu_{t+1})] - \mathbb{E}[f(\mu_t)] \leq \frac{5\eta s K L^2 \mathbb{E}[\Phi_t]}{n(n+1)} + \left(\frac{4sL^2\eta^3 K^3}{n+1} + \frac{2s^2 K \eta^2 L}{(n+1)^2}\right)(\sigma^2 + 2KG^2)$$
$$+ \frac{(R^2+7)^2\gamma^2 L}{(n+1)^2} + \left(\frac{-3\eta s H}{4(n+1)} + \frac{8B^2 L^2 \eta^3 s K^3}{n+1}\right.$$
$$\left. + \frac{4B^2 s^2 K^2 L \eta^2}{(n+1)^2}\right)\mathbb{E}\|\nabla f(\mu_t)\|^2$$

by summing the above inequality for $t = 0$ to $t = T-1$, we get that

$$\mathbb{E}[f(\mu_T)] - f(\mu_0) \leq \frac{5\eta s K L^2}{n(n+1)}\sum_{t=0}^{T-1}\mathbb{E}[\Phi_t] + \left(\frac{4sL^2\eta^3 K^3}{n+1} + \frac{2s^2 K \eta^2 L}{(n+1)^2}\right)(\sigma^2 + 2KG^2)$$
$$+ \left(\frac{-3\eta s H}{4(n+1)} + \frac{8B^2 L^2 \eta^3 s K^3}{n+1} + \frac{4B^2 s^2 K^2 L \eta^2}{(n+1)^2}\right)\sum_{t=0}^{T-1}\mathbb{E}\|\nabla f(\mu_t)\|^2$$
$$+ \frac{(R^2+7)^2\gamma^2 LT}{(n+1)^2}$$

Further, we use Lemma B.5:

$$\mathbb{E}[f(\mu_T)] - f(\mu_0) \leq \frac{5\eta s K L^2}{n(n+1)}\left(80Tn^2(R^2+7)^2\gamma^2 + 80Tn^2 s K\eta^2(\sigma^2 + 2KG^2)\right.$$
$$\left. + 160B^2 n^2 s K^2 \eta^2 \sum_{t=0}^{T-1}\mathbb{E}\|\nabla f(\mu_t)\|^2\right)$$
$$+ \left(\frac{4sL^2\eta^3 K^3}{n+1} + \frac{2s^2 K \eta^2 L}{(n+1)^2}\right)(\sigma^2 + 2KG^2) + \frac{(R^2+7)^2\gamma^2 LT}{(n+1)^2} +$$
$$\left(\frac{-3\eta s H}{4(n+1)} + \frac{8B^2 L^2 \eta^3 s K^3}{n+1} + \frac{4B^2 s^2 K^2 L \eta^2}{(n+1)^2}\right)\sum_{t=0}^{T-1}\mathbb{E}\|\nabla f(\mu_t)\|^2$$
$$\leq \frac{400\eta s n K L^2 T(R^2+7)^2\gamma^2}{n+1} + \frac{404Tns^2 K^2 L^2 \eta^3(\sigma^2 + 2KG^2)}{n+1}$$
$$+ \frac{3s^2 K \eta^2 LT(\sigma^2 + 2KG^2)}{(n+1)^2} + \frac{(R^2+7)^2\gamma^2 LT}{(n+1)^2}$$
$$+ \left(\frac{-3\eta s H}{4(n+1)} + \frac{8B^2 L^2 \eta^3 s K^3}{n+1}\right.$$
$$\left. + \frac{4B^2 s^2 K^2 L \eta^2}{(n+1)^2} + \frac{800B^2 n s^2 K^3 \eta^3 L^2}{n+1}\right)\sum_{t=0}^{T-1}\mathbb{E}\|\nabla f(\mu_t)\|^2$$

by assuming $\eta < \frac{1}{100B\sqrt{nsk^2 L}}$ we get:

$$\mathbb{E}[f(\mu_T)] - f(\mu_0) \leq \frac{400\eta s n K L^2 T(R^2+7)^2\gamma^2}{n+1} +$$
$$+ \left(\frac{3s^2 K \eta^2 LT}{(n+1)^2} + \frac{404Tns^2 K^2 L^2 \eta^3}{n+1}\right)(\sigma^2 + 2KG^2) + \frac{(R^2+7)^2\gamma^2 LT}{(n+1)^2}$$
$$+ \frac{-\eta s H}{2(n+1)}\sum_{t=0}^{T-1}\mathbb{E}\|\nabla f(\mu_t)\|^2$$

Next, we regroup terms, multiply both sides by $\frac{2(n+1)}{\eta s H T}$ and use the fact that $f(\mu_T) \geq f_*$:

$$\frac{1}{T}\sum_{t=0}^{T-1}\mathbb{E}\|\nabla f(\mu_t)\|^2 \leq \frac{2(n+1)(f(\mu_0) - f_*)}{sH\eta T} + \frac{800nKL^2(R^2+7)^2\gamma^2}{H} +$$

$$+ \left( \frac{6sK\eta L}{H(n+1)} + \frac{808nsK^2L^2\eta^2}{H} \right)(\sigma^2 + 2KG^2) + \frac{2(R^2+7)^2\gamma^2 L}{(n+1)sH\eta}$$

Finally, we set $\eta = \frac{n+1}{sH\sqrt{T}}$:

$$\frac{1}{T}\sum_{t=0}^{T-1}\mathbb{E}\|\nabla f(\mu_t)\|^2 \leq \frac{2(f(\mu_0)-f_*)}{\sqrt{T}} + \frac{800nKL^2(R^2+7)^2\gamma^2}{H} + \frac{6KL(\sigma^2+2KG^2)}{H^2\sqrt{T}} \quad (6)$$

$$+ \frac{808n(n+1)^2K^2L^2}{sH^3T}(\sigma^2+2KG^2) + \frac{2(R^2+7)^2\gamma^2 L\sqrt{T}}{(n+1)^2sH} \quad (7)$$

$\square$

**Lemma B.8.** *For quantization parameters* $(R^2+7)^2\gamma^2 = \frac{(n+1)^2}{s^2H^2T}(\sigma^2 + 2KG^2 + \frac{f(\mu_0)-f_*}{L})$ *we have:*

$$\frac{1}{T}\sum_{t=0}^{T-1}\mathbb{E}\|\nabla f(\mu_t)\|^2 \leq \frac{5(f(\mu_0)-f_*)}{\sqrt{T}} + \frac{8KL(\sigma^2+2KG^2)}{H^2\sqrt{T}}$$

$$+ \frac{1608n(n+1)^2K^2L^2(\sigma^2+2KG^2)}{sH^3T} + \frac{800n(n+1)^2KL(f(\mu_0)-f_*)}{s^2H^3T}$$

*Proof.*

$$\frac{1}{T}\sum_{t=0}^{T-1}\mathbb{E}\|\nabla f(\mu_t)\|^2 \leq \frac{2(f(\mu_0)-f_*)}{\sqrt{T}} + \frac{800nKL^2(R^2+7)^2\gamma^2}{H} + \frac{6KL(\sigma^2+2KG^2)}{H^2\sqrt{T}}$$

$$+ \frac{808n(n+1)^2K^2L^2}{sH^3T}(\sigma^2+2KG^2) + \frac{2(R^2+7)^2\gamma^2 L\sqrt{T}}{(n+1)^2sH}$$

$$= \frac{2(f(\mu_0)-f_*)}{\sqrt{T}} + \frac{800nKL^2(n+1)^2}{s^2H^3T}(\sigma^2+2KG^2+\frac{f(\mu_0)-f_*}{L})$$

$$+ \frac{6KL(\sigma^2+2KG^2)}{H^2\sqrt{T}} + \frac{808n(n+1)^2K^2L^2}{sH^3T}(\sigma^2+2KG^2)$$

$$+ \frac{2L}{s^2H^2\sqrt{T}}(\sigma^2+2KG^2+\frac{f(\mu_0)-f_*}{L})$$

$$\leq \frac{5(f(\mu_0)-f_*)}{\sqrt{T}} + \frac{8KL(\sigma^2+2KG^2)}{H^2\sqrt{T}}$$

$$+ \frac{1608n(n+1)^2K^2L^2(\sigma^2+2KG^2)}{sH^3T}$$

$$+ \frac{800n(n+1)^2KL(f(\mu_0)-f_*)}{s^2H^3T}$$

$\square$

**Lemma B.9.** *We have:*

$$5s\sum_{t=0}^{T-1}\mathbb{E}[\Phi_t] + 3\eta^2\sum_{t=0}^{T-1}\sum_i \mathbb{E}\|\widetilde{h}_{i,t}\|^2 \leq 1000Tn^3s(R^2+7)^2\gamma^2$$

$$+ 10000B^2n^3s^2K^2LT(R^2+7)^2\gamma^2$$

*Proof.*

$$5s\sum_{t=0}^{T-1}\mathbb{E}[\Phi_t] + 3\eta^2\sum_{t=0}^{T-1}\sum_i \mathbb{E}\|\widetilde{h}_{i,t}\|^2$$

$$\leq 5s\sum_{t=0}^{T-1}\mathbb{E}[\Phi_t] + 3\eta^2\sum_{t=0}^{T-1}\left(2nK(\sigma^2+2KG^2) + 8L^2K^2\mathbb{E}[\Phi_t] + 4nK^2B^2\mathbb{E}\|\nabla f(\mu_t)\|^2\right)$$

$$\leq 5s \sum_{t=0}^{T-1} \mathbb{E}[\Phi_t] + 6nT\eta^2 K(\sigma^2 + 2KG^2) + 24\eta^2 L^2 K^2 \sum_{t=0}^{T-1} \mathbb{E}[\Phi_t]$$

$$+ 12nB^2\eta^2 K^2 \sum_{t=0}^{T-1} \mathbb{E}\|\nabla f(\mu_t)\|^2$$

$$\leq 6s \sum_{t=0}^{T-1} \mathbb{E}[\Phi_t] + 6nT\eta^2 K(\sigma^2 + 2KG^2) + 12B^2 n\eta^2 K^2 \sum_{t=0}^{T-1} \mathbb{E}\|\nabla f(\mu_t)\|^2$$

$$\leq 6s \big( 80Tn^2(R^2+7)^2\gamma^2 + 80Tn^2 sK\eta^2(\sigma^2 + 2KG^2)$$

$$+ 160B^2 n^2 sK^2 \eta^2 \sum_{t=0}^{T-1} \mathbb{E}\|\nabla f(\mu_t)\|^2 \big)$$

$$+ 6nT\eta^2 K(\sigma^2 + 2KG^2) + 12B^2 n\eta^2 K^2 \sum_{t=0}^{T-1} \mathbb{E}\|\nabla f(\mu_t)\|^2$$

$$\leq 480Tn^2 s(R^2+7)^2\gamma^2 + (480Tn^2 s^2 K\eta^2 + 6nT\eta^2 K)(\sigma^2 + 2KG^2)$$

$$+ (960n^2 s^2 K^2 B^2 \eta^2 + 12B^2 n\eta^2 K^2) \sum_{t=0}^{T-1} \mathbb{E}\|\nabla f(\mu_t)\|^2$$

$$\leq 480Tn^2 s(R^2+7)^2\gamma^2 + 486Tn^2 s^2 K\eta^2(\sigma^2 + 2KG^2)$$

$$+ 1000B^2 n^2 s^2 K^2 \eta^2 \sum_{t=0}^{T-1} \mathbb{E}\|\nabla f(\mu_t)\|^2$$

$$\leq 480Tn^2 s(R^2+7)^2\gamma^2 + 486Tn^2 s^2 K\eta^2(\sigma^2 + 2KG^2)$$

$$+ 1000B^2 n^2 s^2 K^2 \eta^2 \Big( \frac{2(n+1)(f(\mu_0) - f_*)}{sH\eta} + \frac{800TnKL^2(R^2+7)^2\gamma^2}{H} +$$

$$+ \big( \frac{6TsK\eta L}{H(n+1)} + \frac{808TnsK^2 L^2 \eta^2}{H} \big)(\sigma^2 + 2KG^2) + \frac{2T(R^2+7)^2\gamma^2 L}{(n+1)sH\eta} \Big)$$

$$\leq 480Tn^2 s(R^2+7)^2\gamma^2 + 486Tn^2 s^2 K\eta^2(\sigma^2 + 2KG^2)$$

$$+ \frac{2000B^2 n^2(n+1)sK^2 \eta(f(\mu_0) - f_*)}{H} + \frac{800000TB^2 n^3 s^2 K^3 \eta^2 L^2(R^2+7)^2\gamma^2}{H} +$$

$$+ \big( \frac{6000TB^2 n^2 s^3 K^3 \eta^3 L}{H(n+1)} + \frac{808000TB^2 n^3 s^3 K^4 \eta^4 L^2}{H} \big)(\sigma^2 + 2KG^2)$$

$$+ \frac{4000TB^2 nsK^2 \eta L(R^2+7)^2\gamma^2}{H}$$

$$\leq 1000Tn^3 s(R^2+7)^2\gamma^2 + \frac{4000B^2 n^3 sK^2(n+1)}{\sqrt{T}}(f(\mu_0) - f_*)$$

$$+ \frac{10000Tn^3(n+1)^2 s^2 K}{T}(\sigma^2 + 2KG^2)$$

$$\leq 1000Tn^3 s(R^2+7)^2\gamma^2 + 10000B^2 n^3 s^2 K^2 LT(R^2+7)^2\gamma^2$$

$\square$

**Lemma 4.8.** *Let* $T \geq O(n^3)$, *then for quantization parameters* $R = 2 + T^{\frac{3}{d}}$ *and* $\gamma^2 = \frac{(n+1)^2(\sigma^2 + 2KG^2 + f(\mu_0) - f_*)}{s^2 H^2 T(R^2+7)^2}$ *we have that the probability of quantization never failing during the entire run of the Algorithm 1 is at least* $1 - O\left(\frac{1}{T}\right)$.

*Proof.* Let $\mathcal{L}_t$ be the event that quantization does not fail during step $t$. Our goal is to show that $Pr[\cup_{t=1}^T \mathcal{L}_t] \geq 1 - O\left(\frac{1}{T}\right)$. In order to do this, we first prove that $Pr[\neg\mathcal{L}_{t+1}|\mathcal{L}_1, \mathcal{L}_2, ..., \mathcal{L}_t] \leq O\left(\frac{1}{T^2}\right)$ (O is with respect to $T$ here).

We need need to lower bound probability that :

$$\forall i \in S : \|X_t - X_t^i\|^2 \leq (R^{R^d}\gamma)^2 \tag{8}$$

$$\|X_t - (X_t^i - \eta\widetilde{h}_{i,t})\|^2 \leq (R^{R^d}\gamma)^2 \tag{9}$$

$$\|X_t - X_t^i\|^2 = O\left(\frac{\gamma^2(poly(T))^2}{R^2}\right) \tag{10}$$

$$\|X_t - (X_t^i - \eta\widetilde{h}_{i,t})\|^2 = O\left(\frac{\gamma^2(poly(T))^2}{R^2}\right) \tag{11}$$

We would like to point out that these conditions are necessary for decoding to succeed, we ignore encoding since it will be counted when someone will try to decode it. Since, $R = 2 + T^{\frac{3}{d}}$ this means that $(R^{R^d})^2 \geq 2^{2T^3} \geq T^{30}$, for large enough $T$. Hence, it is suffices to upper bound the probability that $\sum_{i\in S}\|X_t - X_t^i\|^2 + \sum_{i\in S}\|X_t - (X_t^i - \eta\widetilde{h}_{i,t})\|^2 \geq T^{30}\gamma^2$. To prove this, we have:

$$\sum_{i\in S}\|X_t - X_t^i\|^2 + \sum_{i\in S}\|X_t - (X_t^i - \eta\widetilde{h}_{i,t})\|^2 \leq \sum_{i\in S}(5\|X_t - \mu_t\|^2$$
$$+ 5\|\mu_t - X_t^i\|^2 + 3\eta^2\|\widetilde{h}_{i,t}\|^2) \leq 5s\Phi_t + 3\eta^2\sum_i\|\widetilde{h}_{i,t}\|^2$$

Now, we use Markov's inequality, and Lemma B.9:

$$Pr[5s\Phi_t + 3\eta^2\|\widetilde{h}_{i,t}\|^2 \geq T^{30}\gamma^2|\mathcal{L}_1, \mathcal{L}_2, ..., \mathcal{L}_t] \leq \frac{\mathbb{E}[5s\Phi_t + 3\eta^2\sum_i\|\widetilde{h}_{i,t}\|^2|\mathcal{L}_1, \mathcal{L}_2, ..., \mathcal{L}_t]}{T^{30}\gamma^2}$$

$$\leq \frac{1000Tn^3s(R^2+7)^2\gamma^2 + 10000B^2n^3s^2K^2LT(R^2+7)^2\gamma^2}{T^{30}\gamma^2} \leq O(\frac{1}{T^2})$$

Thus, the failure probability due to the models not being close enough for quantization to be applied is at most $O\left(\frac{1}{T^2}\right)$. Conditioned on the event that $\|X_t - X_t^i\|$ and $\|X_t - (X_t^i - \eta\widetilde{h}_{i,t})\|$ are upper bounded by $T^{15}\gamma$ (This is what we actually lower bounded the probability for using Markov), we get that the probability of quantization algorithm failing is at most

$$\sum_{i\in S}\log\log(\frac{1}{\gamma}\|X_t - X_t^i\|) \cdot O(R^{-d})$$

$$+ \sum_{i\in S}\log\log(\frac{1}{\gamma}\|X_t - (X_t^i - \eta\widetilde{h}_{i,t})\|) \cdot O(R^{-d})$$

$$\leq O\left(\frac{s\log\log T}{T^3}\right) \leq O\left(\frac{1}{T^2}\right).$$

By the law of total probability (to remove conditioning) and the union bound we get that the total probability of failure, either due to not being able to apply quantization or by failure of quantization algorithm itself is at most $O\left(\frac{1}{T^2}\right)$. Finally we use chain rule to get that

$$Pr[\cup_{t=1}^T\mathcal{L}_t] = \prod_{t=1}^T Pr[\mathcal{L}_t| \cup_{s=0}^{t-1}\mathcal{L}_s] = \prod_{t=1}^T\left(1 - Pr[\neg\mathcal{L}_t| \cup_{s=0}^{t-1}\mathcal{L}_s]\right)$$

$$\geq 1 - \sum_{t=1}^T Pr[\neg\mathcal{L}_t| \cup_{s=0}^{t-1}\mathcal{L}_s] \geq 1 - O\left(\frac{1}{T}\right).$$

$\square$

**Lemma 4.9.** *Let $T \geq O(n^3)$, then for quantization parameters $R = 2 + T^{\frac{3}{d}}$ and $\gamma^2 = \frac{\eta^2}{(R^2+7)^2}(\sigma^2 + 2KG^2 + \frac{f(\mu_0)-f_*}{L})$ we have that the expected number of bits used by Algorithm 1 per communication is $O(d\log(n) + \log(T))$.*

*Proof.* At step $t+1$, by Corollary 4.3, we know that the total number of bits used is at most

$$\sum_{i\in S} O\Big(d\log(\frac{R}{\gamma}\|X_t^i - X_t\|)\Big) + O\Big(d\log(\frac{R}{\gamma}\|X_t - (X_t^i - \eta\widetilde{h}_{i,t})\|)\Big)$$

By taking the randomness of agent interaction at step $t+1$ into the account, we get that the expected number of bits used is at most:

$$\sum_S \frac{1}{\binom{n}{s}} \sum_{i\in S} \left( O\Big(d\log(\frac{R}{\gamma}\|X_t^i - X_t\|)\Big) + O\Big(d\log(\frac{R}{\gamma}\|X_t - (X_t^i - \eta\widetilde{h}_{i,t})\|)\Big) \right)$$

$$= \sum_i \frac{s}{n} \left( O\Big(d\log(\frac{R}{\gamma}\|X_t^i - X_t\|)\Big) + O\Big(d\log(\frac{R}{\gamma}\|X_t - (X_t^i - \eta\widetilde{h}_{i,t})\|)\Big) \right)$$

$$=\leq \sum_i \frac{s}{n} \left( O\Big(d\log(\frac{R^2}{\gamma^2}\|X_t^i - X_t\|^2)\Big) + O\Big(d\log(\frac{R^2}{\gamma^2}\|X_t - (X_t^i - \eta\widetilde{h}_{i,t})\|^2)\Big) \right)$$

$$\overset{Jensen}{\leq} s\left( O\Big(d\log(\frac{R^2}{\gamma^2}\sum_i \frac{1}{n}(\|X_t^i - X_t\|^2 + \|X_t - (X_t^i - \eta\widetilde{h}_{i,t})\|^2))\Big) \right)$$

$$\leq s\left( O\Big(d\log(\frac{R^2}{\gamma^2}\sum_i \frac{1}{n}(\|X_t - \mu_t\|^2 + \|X_t^i - \mu_t\|^2 + \eta^2\|\widetilde{h}_{i,t}\|^2))\Big) \right)$$

$$\leq s\left( O\Big(d\log(\frac{R^2}{\gamma^2}(\Phi_t + \frac{\eta^2}{n}\sum_i \|\widetilde{h}_{i,t}\|^2))\Big) \right)$$

So the expected number of bits per communication in all rounds is at most:

$$\frac{1}{sT}\sum_{t=0}^{T-1} s\left( O\Big(d\log(\frac{R^2}{\gamma^2}(\Phi_t + \frac{\eta^2}{n}\sum_i \|\widetilde{h}_{i,t}\|^2))\Big) \right)$$

$$\leq \left( O\Big(d\log(\frac{R^2}{\gamma^2}(\frac{1}{T}\sum_{t=0}^{T-1}\Phi_t + \frac{1}{T}\sum_{t=0}^{T-1}\frac{\eta^2}{n}\sum_i \|\widetilde{h}_{i,t}\|^2))\Big) \right)$$

Next, By Jensen inequality and Lemma B.9, We get that the expected number of bits used is at most,

$$O\Big(d\mathbb{E}\Big[\log(\frac{R^2}{\gamma^2}(\frac{1}{T}\sum_{t=0}^{T-1}\Phi_t + \frac{1}{T}\sum_{t=0}^{T-1}\frac{\eta^2}{n}\sum_i \|\widetilde{h}_{i,t}\|^2))\Big]\Big)$$

$$\overset{Jensen}{\leq} O\Big(d\log(\frac{R^2}{\gamma^2}(\frac{1}{T}\sum_{t=0}^{T-1}\mathbb{E}[\Phi_t] + \frac{1}{T}\sum_{t=0}^{T-1}\frac{\eta^2}{n}\sum_i \mathbb{E}\|\widetilde{h}_{i,t}\|^2))\Big)$$

$$\leq O\Big(d\log(\frac{R^2}{\gamma^2}(\frac{1}{T}(1000Tn^3s(R^2+7)^2\gamma^2 + 10000B^2n^3s^2K^2LT(R^2+7)^2\gamma^2)))\Big)$$

$$\leq O\Big(d\log(R^2(1000n^3s(R^2+7)^2 + 10000B^2n^3s^2K^2L(R^2+7)^2))\Big) = O(d\log(n) + \log(T))$$

$\square$

**Theorem 4.2.** *Assume the total number of steps $T \geq \Omega(n^3)$, the learning rate $\eta = \frac{n+1}{sH\sqrt{T}}$, and quantization parameters $R = 2 + T^{\frac{3}{d}}$ and $\gamma^2 = \frac{\eta^2}{(R^2+7)^2}\left(\sigma^2 + 2KG^2 + \frac{f(\mu_0)-f_*}{L}\right)$. Let $H > 0$ be the expected number of local steps already performed by a client when interacting with the server. Then, with probability at least $1 - O(\frac{1}{T})$ we have that Algorithm 1 converges at the following rate*

$$\frac{1}{T}\sum_{t=0}^{T-1}\mathbb{E}\|\nabla f(\mu_t)\|^2 \leq \frac{5(f(\mu_0)-f_*)}{\sqrt{T}} + \frac{8KL(\sigma^2 + 2KG^2)}{H^2\sqrt{T}} + O\left(\frac{n^3KL^2(\sigma^2 + 2KG^2)}{sH^3T}\right)$$

*and uses $O(sT(d\log n + \log T))$ expected communication bits in total.*

*Proof.* The proof simply follows from combining Lemmas B.8, 4.8 and 4.9 $\square$

**Lemma B.10.** *For the convergence of the server, we have:*

$$\frac{1}{T}\sum_{t=0}^{T-1}\mathbb{E}\|\nabla f(X_t)\|^2 \leq \frac{15(f(\mu_0) - f_*)}{\sqrt{T}} + \frac{24KL(\sigma^2 + 2KG^2)}{H^2\sqrt{T}}$$
$$+ \left(\frac{4824n(n+1)^2K^2L^2}{sH^3T} + \frac{320n^2(n+1)^2KL^2}{sH^2T}\right)(\sigma^2 + 2KG^2)$$
$$+ \left(\frac{2400n(n+1)^2KL}{s^2H^3T} + \frac{160n^2(n+1)^2L^2}{s^2H^2T}\right)(f(\mu_0) - f_*)$$

*Proof.*

$$\frac{1}{T}\sum_{t=0}^{T-1}\mathbb{E}\|\nabla f(X_t)\|^2 \leq \frac{1}{T}\sum_{t=0}^{T-1}\mathbb{E}\|\nabla f(X_t) - \nabla f(\mu_t) + \nabla f(\mu_t)\|^2$$

$$\leq \frac{2}{T}\sum_{t=0}^{T-1}\mathbb{E}\|\nabla f(X_t) - \nabla f(\mu_t)\|^2 + \frac{2}{T}\sum_{t=0}^{T-1}\|\nabla f(\mu_t)\|^2$$

$$\leq \frac{2L^2}{T}\sum_{t=0}^{T-1}\mathbb{E}\|X_t - \mu_t\|^2 + \frac{2}{T}\sum_{t=0}^{T-1}\|\nabla f(\mu_t)\|^2$$

$$\leq \frac{2L^2}{T}\sum_{t=0}^{T-1}\mathbb{E}[\Phi_t] + \frac{2}{T}\sum_{t=0}^{T-1}\|\nabla f(\mu_t)\|^2$$

$$\leq 2L^2\big(80n^2(R^2+7)^2\gamma^2 + 80n^2sK\eta^2(\sigma^2 + 2KG^2)$$
$$+ 160B^2n^2sK^2\eta^2\frac{1}{T}\sum_{t=0}^{T-1}\mathbb{E}\|\nabla f(\mu_t)\|^2\big) + \frac{2}{T}\sum_{t=0}^{T-1}\|\nabla f(\mu_t)\|^2$$

$$\leq 160n^2L^2(R^2+7)^2\gamma^2 + 160n^2sKL^2\eta^2(\sigma^2 + 2KG^2) + \frac{3}{T}\sum_{t=0}^{T-1}\|\nabla f(\mu_t)\|^2$$

$$\leq 160n^2L^2(R^2+7)^2\gamma^2 + 160n^2sKL^2\eta^2(\sigma^2 + 2KG^2) + \frac{15(f(\mu_0) - f_*)}{\sqrt{T}} + \frac{24KL(\sigma^2 + 2KG^2)}{H^2\sqrt{T}}$$
$$+ \frac{4824n(n+1)^2K^2L^2(\sigma^2 + 2KG^2)}{sH^3T} + \frac{2400n(n+1)^2KL(f(\mu_0) - f_*)}{s^2H^3T}$$

$$\leq 160n^2L^2\eta^2(\sigma^2 + 2KG^2 + \frac{f(\mu_0) - f_*}{L}) + 160n^2sKL^2\eta^2(\sigma^2 + 2KG^2) + \frac{15(f(\mu_0) - f_*)}{\sqrt{T}}$$
$$+ \frac{24KL(\sigma^2 + 2KG^2)}{H^2\sqrt{T}} + \frac{4824n(n+1)^2K^2L^2(\sigma^2 + 2KG^2)}{sH^3T}$$
$$+ \frac{2400n(n+1)^2KL(f(\mu_0) - f_*)}{s^2H^3T}$$

$$\leq \frac{15(f(\mu_0) - f_*)}{\sqrt{T}} + \frac{24KL(\sigma^2 + 2KG^2)}{H^2\sqrt{T}}$$
$$+ \left(\frac{4824n(n+1)^2K^2L^2}{sH^3T} + \frac{320n^2(n+1)^2KL^2}{sH^2T}\right)(\sigma^2 + 2KG^2)$$
$$+ \left(\frac{2400n(n+1)^2KL}{s^2H^3T} + \frac{160n^2(n+1)^2L^2}{s^2H^2T}\right)(f(\mu_0) - f_*)$$

□

Finally, the **proof of Corollary 4.3** follows from combining Lemmas B.10, 4.8 and 4.9

