# OpenReview forum: "QuAFL: Federated Averaging Made Asynchronous and Communication-Efficient"
_ICLR.cc/2023/Conference — Submitted to ICLR 2023_

### Official Review · Reviewer_ea7y · 2022-10-25

**Confidence:** 4
**Clarity, Quality, Novelty And Reproducibility:** See above.
**Correctness:** 2
**Technical Novelty And Significance:** 3
**Empirical Novelty And Significance:** 2
**Recommendation:** 3

**Strength And Weaknesses:**

S1. The FL problem is important and may interest broad audience of ICLR. The asynchronous model seems novel and meaningful in practice.

S2. The delay model considered here more flexible than the classical model with bounded delays, since here the delay can be unbounded in worst case.

---------------------
W1. In Algorithm 1 line 5, $Y_i$ is used without definition. I understand the global aggregation rule, but for the local update, why is the global model weighed by $1/(s+1)$? In my understanding, if there are very slow clients, then at local synchronization we should put a large weight on the global model to help the slow clients catch up with the global model. Say, if we have million of clients and $s=n$, then all the clients basically only update on their own data and do not synchronize with the server ($1/(s+1)$ is negligibly small). I doubt that the algorithm still works in this case. Could you please provide more intuition on this weight?

W2. There are some overstatements regarding the theoretical results. The $O(1/\sqrt{T})$ rate in Corollary 4.3 does not match the best convergence rate of FedAvg in Karimireddy et al. (2020). In that paper, the authors showed that the convergence rate of FedAvg can be $1/\sqrt{Ts}$.

W3. In Corollary 4.3 the learning rate $\eta$ increases when $s$ decreases. This is counter-intuitive since typically, when there is fewer participating clients, the variance of gradient estimation gets larger and one need to use smaller learning rates. Additionally in this asynchronous case, smaller $s$ means more delays, which implies larger gradient error caused by staleness. Thus, I'm not very sure if this learning rate schedule is a correct choice.

W4. The experimental evaluation is insufficient. Most results are ablation study and the proposed method is only compared with standard FedAvg in one figure. There are more compression methods other than the lattice quantization used in QUAFL which can be tested. There  might also be other aggregation strategies such as the one I mentioned above, directly updating the local model with the global model when interacted. More results should be provided to justify the unique advantage of the new update rule and quantizer, compared with some standard methods.

------------------------------------------------------------------------------------------
Post-rebuttal:

I would like to thank the authors for the reply and adding more experiments. I appreciate the additional results on QSGD and FedBuff. However, the rebuttal does not fully address my concerns.

1. While there is one figure in the supplemental file that shows the result with $n=300$, the number of active clients $s$ is only 30, which is still quite small. Also, it seems that the experiments are simulated on iid clients as described in Section A.2. Thus, this is not very convincing to resolve my question on the vanishing weight of the global model in the local model update with large $s$ asymptotically. When $s$ is very large, the local models train on their own data without communicating with others, and the global model simply takes the average of the local models. In my understanding, this should not work with non-iid clients. In addition, I think the "optimal approach" you mentioned is exactly the standard averaging strategy in which the global weight equals 1 instead of $1/(s+1)$.

2. Thanks for the explanation and please add some discussion in the paper that your rate matches FedAvg under some specific parameter settings and in the wall-clock sense.

3. In the scaffold paper, they consider two learning rates (global $\eta_g$ and local $\eta_l$) and the effect of $s$ is "balanced out". In your method you only use one learning rate which increases with smaller $s$. Thus, this still looks unnatural and incorrect to me.

I appreciate the efforts in adding more experiments and baselines. Yet I still have some concerns on the correctness of the algorithm and analysis (and the iid experimental setting). So I will keep my score.

**Summary Of The Paper:**

The paper studies asynchorous federated learning with communication compression called QUAFL. In this model, The server randomly pick clients whose gradient might be stale, and clients are assumed to have positive expected number of local steps each time it interacts with the server.  A new model aggregation rule is designed in the paper, with convergence analysis. Experiments illustrate the effectiveness of the method.

**Summary Of The Review:**

The paper studies an interesting problem in asynchronous FL. However, I have a few concerns about the algorithm and theoretical analysis. More clarification and experiments might be needed to address this issues. Moreover, the experiments, which mostly consist of ablation studies, are insufficient to show the benefit of the proposed strategies. Thus, I think the paper does not reach the high bar of ICLR.

---

> ### Author Response · Authors · 2022-11-16
> **Review Response Part 1**
>
> Thank you for your questions. We address them below in detail, in two parts:
>
> > In Algorithm 1 line 5, Yi  is used without definition.
>
> We define Y_i explicitly in the algorithm textual description, i.e. the first paragraph on page 5, where both the client-side and the server-side interactions are considered.
> Please note that Y_i is defined implicitly as the client’s message in line 5, and its exact contents (in terms of the client’s local variables, not defined at line 5) are defined on line 2 of InteractWithServer. In the revision, we add a comment on line 5 to clarify this.
>
> > I understand the global aggregation rule, but for the local update, why is the global model weighed by 1/(s+1)? In my understanding, if there are very slow clients, then at local synchronization we should put a large weight on the global model to help the slow clients catch up with the global model. Say, if we have million of clients and s=n, then all the clients basically only update on their own data and do not synchronize with the server (1/(s+1) is negligibly small). I doubt that the algorithm still works in this case. Could you please provide more intuition on this weight?
>
> We touched upon this in the discussion paragraph on page 5, and in Figure 3.
> Specifically, this precise formulation of the update is important theoretically, since this is the only weighting which maintains overall “balance” in our potential function \Gamma, allowing us to show super-martingale behavior for it. Experimentally, we show in Figure 3 (page 8) that this weighting provides the fastest convergence among all other basic variants (w.r.t. all 4 combinations of client-side vs server-side averaging). To your question, notice that, if we would consistently put smaller weight on slow clients, then, theoretically, their contribution in the sum loss would be *lower* than for fast clients, leading to optimizing a potentially different objective.
>
> In more detail: indeed, the fraction of the server model in the averaging is $1/(s+1)$. Yet, from the global perspective of the potential \Gamma, this is not the same as ignoring the server’s contribution: the server gets averaged with $s$ clients, and each of them plus the server itself gets a $1/(s+1)$ fraction of the server model. Thus, the local model of the server becomes evenly distributed among $s+1$ participants, as it should be in terms of load balancing. In fact the optimal approach would be to distribute all local models among $s+1$ participants, but unfortunately this will be costly communication-wise and will not work well with asynchrony.
>
> From the practical perspective, notice that we have shown our algorithm to work well even for n = 300 nodes, at which point the server share is fairly small.

---

> ### Author Response · Authors · 2022-11-16
> **Response Part 2**
>
>
> > There are some overstatements regarding the theoretical results. The O(1/T) rate in Corollary 4.3 does not match the best convergence rate of FedAvg in Karimireddy et al. (2020).
>
> Please notice that our precise formulation of the claim is that, we can match the FedAvg bound for s = Theta(n).  Specifically, here we meant that we can achieve O( 1 / \sqrt(Tn) ) convergence in terms of *wall-clock* time, that is, taking execution parallelism into account.
> More precisely, consider the setting where the server samples one (or a constant number) of nodes in each “server step.” In this case, the algorithm at a single step is simply a single ping-pong transmission step between the server and one randomly-chosen client. At the same time, FedAvg with s = Theta(n) would require a synchronous all-to-all between the server and (a majority of) all clients.
>
> Our argument, similar to the one made by [Koloskova et al., NeurIPS 2022]; is that our asynchronous algorithm should be able to do Theta(n) individual transmissions in the same amount of wall-clock time as FedAvg with s = Theta(n), as the communication pattern of FedAvg is synchronous, and so FedAvg must wait for each client to complete its local steps and transmit back the model.
> Thus, we can replace T -> nT when discussing our algorithm’s convergence in terms of wall-clock time, to account for the fact that our asynchronous interactions should happen n times faster than an all-to-all by the server in FedAvg. In this case, our rate asymptotically matches that of FedAvg with s = \Theta(n).
>
> Moreover, we asymptotically match the FedAvg rate for constant s.
> We do acknowledge the reviewer’s clarification request, and will update the discussion to reflect this point.
>
> We cannot match the FedAvg bound in general, w.r.t. the actual number of gradient steps taken. This is because our algorithm is asynchronous: for example, if the server samples s = \sqrt(n) clients per round, then a sampled client at a step is Theta(\sqrt{n}) rounds “behind” the server on average, so our algorithm cannot match a fully-synchronous one, as per known lower bounds on asynchronous SGD.
>
> >  In Corollary 4.3 the learning rate η increases when s decreases.
>
> Indeed we initially also found this counter-intuitive.
> Yet, please note that, the synchronous FedAvg analysis of Scaffold, the LR is also decreasing as s increases (see their Thm. 1).
>
> W.r.t. delays, as you also observed, we do not make any assumptions about the max delays. The only parameter which can be used to encode delays is H. In some settings, we can view H as proportional to n/s:  consider the scenario where in each server round every node makes exactly one local step, hence once interrupted the expected number of local steps performed by a given node is n/s. In that case, our learning rate does not decrease since we have sH=n in the numerator.
> Unfortunately, we are not able to set the learning rate to even larger values, the reason being that we make no assumptions on the delays. Subsequently, the only way we can bound large deviations of the model is by load balancing. A larger learning rate would lead to a larger deviation at a step, which is why we needed to bound our maximum LR.
>
> > Most results are ablation study and the proposed method is only compared with standard FedAvg in one figure. There are more compression methods other than the lattice quantization used in QUAFL which can be tested.
>
> Please note that, in Figure 3, we compared convergence between the QuAFL update, and 3 other update types (w.r.t. server-side and client-side averaging). Clearly, the QuAFL update performs the best among all other choices. Moreover, the other updates are not fully theoretically-justified.  The Appendix contains several additional comparisons relative to variants of FedAvg, or the SGD baseline, in Figures 8,9,10,11,12.
>
> In addition, we emphasize that using a standard quantizer (e.g. stochastic quantization) in an asynchronous algorithm is not theoretically-justified: in this case, we would not be able to prove convergence, since the quantization error could be unboundedly large.
> In general, since there is no other FL algorithm to support both asynchrony and quantization with guarantees, it is not clear to us what we should compare against in a fair manner.
>
> **Nevertheless, in the final revision we present experiments where we compare against FedBuff, a state-of-the-art asynchronous FL algorithm. In addition, we naively use the QSGD stochastic quantizer to compress updates both in QuAFL and in FedBuff. Our results in [Figure 5](https://openreview.net/pdf?id=Rb3RN0maB7F#page=8) and [Figure 16](https://openreview.net/pdf?id=Rb3RN0maB7F#page=16) show that, it is possible for the resulting algorithms to not diverge. However, we observe significantly worse convergence.**
>
> We would be happy to know if we addressed your concerns.

---

### Official Review · Reviewer_ZjQd · 2022-10-26

**Confidence:** 3
**Correctness:** 4
**Technical Novelty And Significance:** 2
**Empirical Novelty And Significance:** 2
**Recommendation:** 5

**Clarity, Quality, Novelty And Reproducibility:**

The paper is well written.

I did not run the experiments myself.



**Strength And Weaknesses:**

Strength:
I think the paper is quite easy to follow and the majority of related works are cited.

Weakness:
It is not clear to me at this point why authors specifically chose Latice quantization and what are the advantages of lattice quantization compared to the rest.
I think it is very important to compare the tightness of the bound with prior work asyncronous FL algorithms such as [A]. What I would like to see is that if your bound of Theorem 4.2 if I do not use quantization $\gamma=0$ how would your result compete with prior studies? Unfortunately, It is not obvious to me, and not sure whether you could improve the bounds in prior bounds.
Another limitation I see is that for both Theorems 4.2 and corollary 4.3 you need to have $T\geq \Omega (n^3)$ and $T\geq \Omega (n^4)$. In the convergence of FedAvg we do not have these constraints and how would the author comment on that?
I think the experiment section is also limited. Authors would want to compare with other Asynchrnous FL algorithms as well. Furthermore, as our ultimate goal is accuracy, Authors should compare the accuracy of their results with FedAvg and other SOTA algorithms.

[A]: Koloskova, Anastasia, Sebastian U. Stich, and Martin Jaggi. "Sharper convergence guarantees for asynchronous SGD for distributed and federated learning." arXiv preprint arXiv:2206.08307 (2022).

Minor comment:
I think the contributions part should be more concise.


**Summary Of The Paper:**

This paper incorporates quantization and asynchronous communication to well known Federated Learning algorithm. The compression scheme is based on lattice quantization. Convergence analysis of the proposed algorithm is provided and empirically verified through a set of experiments.


**Summary Of The Review:**

Please see the comments above!
I will update my score based on the authors feedback.

---

> ### Author Response · Authors · 2022-11-16
> **Review Response Part 1**
>
> Thank you for your very interesting questions! We split our response in two parts:
>
> > It is not clear to me why authors specifically chose Latice quantization and what are the advantages.
>
> The key advantage of the lattice quantizer is that its error does not depend on the norm (L2) of the vector being compressed, but on the norm of the difference between the decoding reference point and the corresponding vector. This simplifies analysis in the FL setting, for two reasons:
>
> * *If the *model* is being transmitted/quantized*, which is the case in some variants of FedAvg, then the quantization error using general quantizers (e.g. stochastic rounding) becomes unbounded.
>
> * *If the *gradients/sum of model updates* are being quantized*, then this usually requires either a second-moment bound on individual gradients, and/or some homogeneity assumption on the local data as well as a deterministic bound on the delay \tau. Notice that this is the case in much of the prior work on communication-compressed FL, as discussed at length in our related work section.
> Notice that this approach would become even more complicated in case we combined *asynchrony* with quantization, as in that case the quantization error would start depending on the max delay.
> (It may be possible to overcome this, using some more complex argument, or by carefully-decreasing stepsizes, but at that point the analysis would probably be more complicated than ours.)
>
> > Another limitation I see is that for both Theorems 4.2 and corollary 4.3 you need to have T≥Ω(n3) and T≥Ω(n4). In the convergence of FedAvg we do not have these constraints and how would the author comment on that?
>
> This is indeed a technical limitation of our potential analysis, as the bound on T is required to ensure that the averaging process between the server and clients “mixes” well. We note that this assumption is not completely unreasonable given that SGD is usually run for a large number of steps in the federated setting. Moreover, our experiments show that, in practice, the algorithm converges well even for low T.
>
> > Authors should compare the accuracy of their results with FedAvg and other SOTA algorithms.
>
> In the revision, we provide full-convergence results for FedAvg vs QuAFL, showing that QuAFL can indeed reach SOTA accuracy in the federated setting, in *faster* wall-clock time than FedAvg. For example, QuAFL reaches >91% accuracy on FMNIST, same as FedAvg/minibatch SGD, but faster in terms of wall-clock time due to leveraging asynchrony.
>
> **Moreover, in the final revision we also present experiments where we compare against FedBuff, a state-of-the-art asynchronous FL algorithm. Our results in [Figure 6](https://openreview.net/pdf?id=Rb3RN0maB7F#page=8) show that QuAFL outperforms FedBuff in terms of convergence vs number of steps. Moreover, we observe significantly worse convergence if naive QSGD quantization is applied to FedBuff, in [Figure 16](https://openreview.net/pdf?id=Rb3RN0maB7F#page=16).**
>
> > Minor comment: I think the contributions part should be more concise.
>
> Thank you, we have addressed this in the revision.

---

> ### Author Response · Authors · 2022-11-16
> **Response Part 2: Comparison with (Koloskova et al., NeurIPS22)**
>
>
> > I think it is very important to compare the tightness of the bound with prior work asyncronous FL algorithms such as [A]. What I would like to see is that if your bound of Theorem 4.2 if I do not use quantization γ=0 how would your result compete with prior studies?
>
> We attempt to provide a detailed discussion here.
>
> Setting comparison:
>
> * First, note that the assumptions and setup between our work and general work on asynchrony, such as [A], are different. Specifically, in work on asynchrony, such as Algorithm 1 in [A], it is common to assume a worst-case/average upper bound on delays ($\tau$ in this reference), and on the worst-case gradient second-moment bound (which can sometimes be relaxed). We do not require either in our work, since we make some basic probabilistic assumptions on the client asynchrony (H > 0, as detailed in the general response), and only work with variance bounds in our analysis.
>
> * Second, as briefly discussed above, our algorithmic and analytic approach is specifically-designed to obtain an algorithm which supports quantization, and in particular certain components could be simplified if this weren’t the case.
>
> *In passing, we note that [A] is concurrent work to ours, as the arXiv posting dates for this work and ours are literally a few days apart.
> Nevertheless, to fully address your question, we attempt to provide a direct technical comparison with the rates of [A] in the case where there is no quantization.*
>
> Technical comparison:
>
> * Our approach allows asynchrony from two sources: 1) the workers perform local updates on an old version of the model than the one at the server, but also 2) to avoid waiting by the server, the worker immediately has to submit their current progress to the server when contacted, even if it is partial.
>
> * By contrast, the model used in [A] would only allow the first type of asynchrony. In the language of [A], our average delay at a client would be $n / s$, as a client gets contacted with probability $s/n$ at a round. **Importantly, we do not specify a maximum delay, which could be unbounded in QuAFL.**
>
> * Reference [A] (in particular Algorithm 2) requires workers to fully complete their local assignment when "polled" by the server. Specifically, the server could submit more than one job on the worker as they receive the model from the server. (For instance, given n consecutive samplings of s clients by the server, there is one client which is chosen by s log n / log log n of them, and a constant fraction may not be chosen at all, w.h.p, assuming constant s.)
>
> * We make a *strictly weaker* assumption in QuAFL: we just assume that the worker performed at least some local work *on average* (H > 0) when contacted. In other words, the client may share the model with < K iterations, hence the current job is interrupted and jobs do not get added on by the server. In practical terms, this weaker assumption on structure is beneficial in order to balance jobs between workers: in the case where local computation times are long, in the algorithm of [A], the server could become stuck waiting on a worker that is sampled multiple times in succession.
>
> * Under these conditions, we match the SGD/[A] bounds for s = constant, if we apply the same “parallel interactions” argument used in [A]. Specifically, our rate is dominated by the first two right-hand-side terms in Theorem 4.2, the first of which is divided by \sqrt{T}, and the second is divided by K \sqrt{T} for H = K.
> However, as argued in [A], in this case n individual client-server steps can be assumed to occur in parallel, and therefore, in terms of wall-clock time, it is valid to make the substitution T -> nT. Thus, we would obtain an "optimal" rate 1/\sqrt{nT} in terms of wall-clock/parallel time. (We note that, besides [A], this parallelism argument is commonly made in the literature on decentralized SGD.)
>
> We would be very happy to find out if we addressed your concerns.

---

### Official Review · Reviewer_xMTC · 2022-10-31

**Confidence:** 3
**Correctness:** 3
**Technical Novelty And Significance:** 2
**Empirical Novelty And Significance:** 1
**Recommendation:** 3

**Clarity, Quality, Novelty And Reproducibility:**

-Sections 3 and 4 may be revised by the authors. A comprehensive comparison and discussion must be made with prior works in these two sections.
-Could you please elaborate on the connection between Assumption 4 and the bounded heterogeneity assumption?
-This work presents a new analysis for asynchronous federated learning with compressed communications building on lattice-based quantization.


**Strength And Weaknesses:**

Strengths:
-New convergence analysis for asynchronous federated learning
-Incorporating the idea of compressed communication with asynchronous updates

Weaknesses:
-The server algorithm has an underlying assumption (lines 4-6 of Algorithm 1) that the server can allocate separate memory to each client's parameters. To this reviewer's understanding, this is not a proper assumption for cross-device federated learning.
-The update in line 6 of Algorithm 1 requires an additional acknowledgment message from the server and client to ensure that the most updated parameter has arrived at the client. When updates are delayed or communication is lost, this creates an additional burden.
-The assumptions on the compression operator must be stated formally in the paper. Lemma 4.1 does not clearly state the required setup for compression.
-Section 3 presents briefly (informally) the assumptions about asynchronous communications. Asynchrony is one of the main contributions of this work, so it should be stated as a separate formal assumption in Section 4.
-There is a limited comparison with prior works. It is necessary to compare the asynchrony assumption to prior works. As far as this reviewer is concerned, it is unclear how this assumption fits into the literature.
-The benefit of compression in terms of exchanged bits (or other metrics) in the experiments must be presented and discussed. The experiments should include comparisons with other algorithms.


**Summary Of The Paper:**

This paper examines the federated learning problem when updates are made asynchronously and communication is compressed. The authors present a new variant of the classic federated averaging (FedAvg) algorithm, namely QuAFL, that supports both asynchronous communication with compression. In some parameter regimes, they demonstrate that their proposed algorithm can provide similar convergence to FedAvg. The authors show that the algorithm ensures fast convergence for a few standard federated learning tasks on the experimental side.


**Summary Of The Review:**

This paper presents an algorithm for asynchronous federated learning with compressed communications. The authors propose a feedback-based algorithm that reduces communication overhead in the federated framework based on existing research on asynchronous updates and lattice-based quantization. This paper focuses primarily on theoretical studies and analysis. A major area for improvement of the work is the need for adequate comparisons with prior works, both theoretically and experimentally. According to the analysis and experiments, there is no clear evidence that compression reduces communication volume. Additionally, some work is required to improve the presentation of this paper.

---

> ### Author Response · Authors · 2022-11-16
> **Review Response**
>
> Thank you for your feedback!
>
> >  The server algorithm has an underlying assumption (lines 4-6 of Algorithm 1) that the server can allocate separate memory to each client's parameters. To this reviewer's understanding, this is not a proper assumption for cross-device federated learning.
>
> We believe there is a significant misunderstanding on this point: our algorithm only needs the server to be able to receive and store s quantized client messages per round (Enc(Y^i)), one per contacted client. This is obviously required for FedAvg as well, since it receives and averages s messages per round. If memory is an issue, messages can be decoded and averaged one-by-one, using a couple of buffers of size d.
>
> Crucially, we stress that the server does not need to store any information about the clients between rounds: decoding of received messages is performed with respect to the server’s current parameter $X_t$, and the received messages are simply discarded after being integrated into X_{t + 1}.  We will clarify this in the revision.
>
> > The update in line 6 of Algorithm 1 requires an additional acknowledgment message from the server and client to ensure that the most updated parameter has arrived at the client. When updates are delayed or communication is lost, this creates an additional burden.
>
> We wish to emphasise there is no further communication needed between server and client, although in hindsight we see that this could be interpreted from the pseudocode. To clarify:
>
> * Communication is always initiated by the server. Upon a communication step, the server contacts s clients at random, by *directly sending* them its current parameter X_t, in encoded form, as part of its "polling" message.
>
> * When receiving the server’s message, each client directly sends it its current progress at this point (line 2 in InteractWithServer). So the client could send its progress before the K local updates are done.
>
> * It then decodes the server’s message and incorporates it into X^i (line 7)
>
> * **Importantly, the decoding of the server message and the encoding & sending of the client message happen independently: the client does not need to receive the “latest” server message after its own message was sent.**
>
> We thank the reviewer for pointing out this source of potential confusion; we will update the text to clarify it.
>
> > The assumptions on the compression operator must be stated formally in the paper. Lemma 4.1 does not clearly state the required setup for compression.
>
> Thank you for this comment, we will clarify the compression setup. We stress that we do not make any non-standard assumptions on the compressor: the only thing that is not straightforward is the decoding key, which is always specified in terms of local information at the receiver (either server or worker).
>
> > Section 3 presents briefly (informally) the assumptions about asynchronous communications. Asynchrony is one of the main contributions of this work, so it should be stated as a separate formal assumption in Section 4.
>
> Thank you, we will add a formally-stated assumption at the end of Section 4.1.
>
> >  There is a limited comparison with prior works. It is necessary to compare the asynchrony assumption to prior works. As far as this reviewer is concerned, it is unclear how this assumption fits into the literature.
>
> We acknowledge this point, and expand our comparison in the revision. In brief, we are assuming a probabilistic bound on the asynchrony, which requires each client to take a non-zero number of steps *in expectation* between communication steps with the server.
>
> Based on our examination of the literature, this assumption is different than the standard worst-case Hogwild!-type bounds (Recht et al., NeurIPS11, Koloskova et al., NeurIPS22), but not strictly stronger. Specifically, we do not assume a deterministic bound on the delay, but instead assume a probabilistic one, which we only bound in expectation. So our worst-case delay is not fixed. As we clarify in the revision, this is similar with some of the modeling made in the asynchronous SGD literature, specifically (Canelli et al., Math.Prog. 2020; Kungurtsev et al., AAAI21).
> We will update the discussion in the revision to address this question.
>
> > The benefit of compression in terms of exchanged bits (or other metrics) in the experiments must be presented and discussed. The experiments should include comparisons with other algorithms.
>
> Please note that we already included ablations of convergence for the algorithm with different bit counts (Figures 2 and 15). In the revision, we will illustrate convergence directly vs. #bits communicated.
>
> Thank you for your feedback. We would be happy to know if it clarified your concerns.

---

### Official Review · Reviewer_oDyr · 2022-11-01

**Confidence:** 4
**Correctness:** 3
**Technical Novelty And Significance:** 2
**Empirical Novelty And Significance:** 2
**Recommendation:** 5

**Clarity, Quality, Novelty And Reproducibility:**

The theoretical analysis is solid but the conclusion is questionable. The paper is generally well-written.

**Strength And Weaknesses:**

Strength
- The algorithm seems to be very complex but the authors are able to provide a very clean theoretical result.
- I can imagine that analyzing an FL algorithm with both compression and asynchrony must be complicated. The authors successfully manage the complex derivation process.
- Most previous analyses on gradient compression used non-standard assumptions. But the authors are able to remove/replace them with standard ones.

Weakness
- The motivation of this paper is a bit weak. In the introduction, the authors start the paper by saying that both compression and asynchrony are needed for practical FL. This is true. However, both techniques have been studied in previous literature. We already get answers to questions like how to apply asynchronous training in FL, and how to use gradient compression in FL. These two techniques are orthogonal to each other and might be easily combined together. It is unclear to me what are the key algorithmic challenges here. If simply combining them does not work, then the authors need to provide some theoretical or empirical evidence. But I didn't find any. If simply combining asynchrony and compression works, then the authors are supposed to change the introduction and motivate the paper mainly from a theoretical perspective.
- It is unclear to me how the asynchrony works in the proposed algorithm. At each round, the server needs to randomly select $s$ clients. Then, immediately after receiving the server request, the clients need to upload their local models. That means these clients should already start local training at an earlier time. Then, what if some clients never participate in training before? It seems that the authors assume that at any time all clients are performing local training, which is impractical and a waste of resources. Could the author answer the question: how many clients are concurrently performing local training at any given time?
- The theoretical result seems to be weak. In particular, the dominant term in Theorem 4.2 is $1/\sqrt{T}$, which does not depend on the number of all clients $n$ nor the number of sampled clients $s$. This rate is slower than vanilla FedAvg, which gets improved rates when selecting more clients.
- It would be better to provide an update rule for the proposed algorithm so that people can easily get what the algorithm exactly does at each round.

**Summary Of The Paper:**

This paper studies how to use communication compression and asynchronous training together in federated learning (in particular, FedAvg algorithm). The authors showed that most previous papers used non-standard assumptions when analyzing the convergence of FedAvg with gradient compression. In their new analysis, they are able to use all standard assumptions. Also, they introduce new analysis techniques to analyze asynchronous behaviors. At last, experiments validate the theoretical predictions on how the algorithm changes along with its hyper-parameters.

**Summary Of The Review:**

While I appreciate the efforts the authors put into the theoretical analysis, the motivation and some technical details of this paper are not satisfactory.

---

> ### Author Response · Authors · 2022-11-16
> **Response Part 1 (Motivation)**
>
> > The motivation of this paper is a bit weak. In the introduction, the authors start the paper by saying that both compression and asynchrony are needed for practical FL. This is true. However, both techniques have been studied in previous literature. We already get answers to questions like how to apply asynchronous training in FL, and how to use gradient compression in FL.
>
> Thank you for the opportunity to clarify this point!
>
> It is indeed true that both asynchrony and communication compression have been considered in the FL literature, in isolation. However, as we discuss in detail in the related work section, *even for synchronous* communication compression the situation is not yet completely resolved:
> * All references which consider versions of the practical FedAvg with compression require non-standard assumptions in their analysis (e.g. i.i.d. local data or second-moment bounds).
> * References which provide stronger bounds, e.g. MARINA or DASHA, are *strictly synchronous* and require much more complex algorithms (e.g., variance-reduced sub-iterations), which would be hard to implement in practice.
> **By contrast, we present bounds for FedAvg-like algorithms with compression and asynchrony, without any non-standard assumptions like i.i.d. data or gradient second-moment bounds.**
>
>
> > These two techniques are orthogonal to each other and might be easily combined together. It is unclear to me what are the key algorithmic challenges here. If simply combining them does not work, then the authors need to provide some theoretical or empirical evidence. But I didn't find any. If simply combining asynchrony and compression works, then the authors are supposed to change the introduction and motivate the paper mainly from a theoretical perspective.
>
> We agree that the question “do standard techniques just work?” is important. We briefly addressed it in the submission (page 2, paragraph 3), and clarify further here:
>
> * **At the analytical level, we believe applying standard quantizers wouldn’t work directly in the context of asynchronous FL, at least not without a very complex analysis.**
>
>
> Specifically, standard compressors (e.g. stochastic) bound error with respect to the norm of the vector being quantized. In the case of  FedAvg, the norm of the transmitted update would be the norm of either the sum of local gradient updates at the client over several rounds, or of the parameter itself. Therefore, this error cannot be bounded without making non-standard assumptions (e.g. a i.i.d. data / gradient second moment). Usually, circumventing this leads to much more complex algorithms and analyses: please see (Koloskova et al., ICML 2019, Lu and De Sa, ICML 2020) for examples in the different decentralized case.
>
> This is precisely why we employ a non-trivial lattice quantizer, which bounds error with respect to a “pivot” point, which we choose very carefully in our algorithm, and show to be valid in the analysis.
> This discussion is clearly also relevant to asynchronous FedAvg algorithms, since in that case the norm update would be even larger and depend on the max delay (since one would consider updates delayed over several asynchronous rounds). Our analysis circumvents this as well.
>
> *  **We showed practically, in Figure 3, that our new “smoothed” averaging procedure at both client and server is necessary for fast convergence.**
> Specifically, variants without it clearly converge worse in practice.
>
> We hope that this answer further motivates our work, and explains the rationale behind some of our algorithmic and analytic choices. We are not aware of any algorithm which has guarantees in the same setting as QuAFL (quantization + asynchrony), which is why our ablations are with respect to prior algorithms in fixed settings which are roughly comparable.
> We do acknowledge your request for clarification on this point, and address it in the revision.

---

> ### Author Response · Authors · 2022-11-16
> **Response Part 2 (Algorithm and Analysis Clarifications)**
>
> > It is unclear to me how the asynchrony works in the proposed algorithm. [...] It seems that the authors assume that at any time all clients are performing local training.
>
> We believe there is a misunderstanding with respect to our setting.
> 1. Initially, all clients start with a randomly-initialized model. (WLOG, assume it is the same for each client.)
> 2. Then, the server waits for a fixed round timeout, while each client tries to perform K local steps on its model.
> The server then contacts $s$ clients at random.
> There are now two options.
>
> 3a. If the client finishes its $K$ local steps (at its own local pace), then it doesn’t do anything else, and waits to be contacted by the server. This will eventually happen since the server samples randomly.
>
> 3b. If the client gets contacted *before* completing its K local steps, we are assuming that it returns its current progress $Y_i$ , and then restarts the K steps, at its own pace.
>
> None of this assumes that all clients are performing local training all the time (see 3a). Importantly, every node performs at most K steps between interactions with the server, and then is idle. Our only assumption is that, when sampled, the *expected* number of local steps taken by the client is >0. Clearly, this is necessary: if some clients always take zero local steps, their data cannot be taken into account.
>
> > Could the author answer the question: how many clients are concurrently performing local training at any given time?
>
> We are happy to clarify this:
>
> 1. **From the implementation perspective:**
>
> We assume n total clients, out of which s are sampled per round by the server. Each client takes 1 time unit on average per local SGD step, and a client will take a maximum of K local steps until waiting to be contacted by the server. The server contacts clients once every SWT (server waiting time) steps.
>
> Then, on average, a client needs K time to finish its local work, and is sampled once every SWT * (n / s) time steps by the server. The difference SWT * (n / s) - K, if positive, is the “rest” time for the worker.
>
> For example, two settings from our experiments are n = 40, s = 10, K = 10, and SWT = 10 (Figure 7), in which case nodes get contacted on average every 40 steps, out of which 10 (25%) are spent working. Another experiment (Figure 5) considers the same values, but K = 5 and SWT = 10, in which case nodes spend 87.5% of their time being idle. It is evident from these figures that the algorithm works well for these settings.
>
> The choice of SWT is in fact up to the system designer. In our experiments we tended to experiment settings with very low “rest” times for the nodes to test the algorithm (lower SWT) in order to stress-test the algorithm: in this case, don’t always finish their local steps, and so we are more likely to observe asynchronous behavior.
> However, we believe that in practice nodes would spend most of their time being idle (since n / s and SWT would be very large).
>
> 2. **From the analysis perspective:**
>
> We sidestep all these practical issues, since we make much weaker assumptions on the modeling.
> Specifically, the only quantity we need in the analysis is the *average number of local steps completed* by the node when contacted. The only assumption we make about it is that it would be > 0 in expectation.
> In terms of the above practical discussion, this average is (n/s) * SWT, and we are assuming that its expectation is > 0, i.e. that the client is likely to have completed at least one step when contacted by the server.
>
> > The theoretical result seems to be weak. In particular, the dominant term in Theorem 4.2 is 1/T, which does not depend on the number of all clients n nor the number of sampled clients s.
>
> It is indeed the case that our rate does not get a speedup with s when measured w.r.t. the *number of gradient steps*, which is because of the impact of asynchrony. However, our algorithm should (and does) get improved convergence vs FedAvg in terms of wall-clock time.
> For instance, QuAFL with constant s = constant should be able to do Theta(n) individual transmissions in the same wall-clock time as FedAvg with s = Theta(n), as FedAvg must wait for each client to complete its local steps and transmit back the model, whereas QuAFL can communicate with clients in parallel.
> Thus, similar to arguments by [Koloskova et al., NeurIPS 2022]; [Nadiradze et al., NeurIPS 2021], if we measure wall-clock time, we can replace T -> nT when discussing our algorithm’s convergence, to account for the fact that our interactions should happen faster than an all-to-all by the server in FedAvg.
> In this case, the rate would asymptotically match that of FedAvg with s = \Theta(n).
>
> > It would be better to provide an update rule for the proposed algorithm.
>
> We try to address this in the revision.
>
> We would be happy to know if we clarified your concerns.

---

### Author Response · Authors · 2022-11-16
**Response Overview**

**Note: This Overview was updated on Nov 18th to reflect additional experiments present in the final revision.**

We would like to thank the reviewers for their feedback!

We provide detailed individual responses to each review, inline. In addition, we plan to submit a detailed revision very soon, which incorporates all the reviewers’ feedback.
Specifically, our responses address the following general points:

1. **The complexity of combining asynchrony and quantization in the context of FedAvg**

Reviewers asked whether combining orthogonal techniques for asynchrony and quantization would not work directly.
We believe that this is definitely not the case, and for this provide the following arguments:

* As discussed in detail in our Related Work section, even for *synchronous* quantization of FedAvg-like algorithms, existing analyses require strong assumptions, such as i.i.d. local data, or gradient second-moment bound. Our analysis does not require such assumptions: we discuss the rate we obtain in the synchronous case in the response to Reviewer ZjQd.

* In addition, as discussed in detail in the response to Reviewers oDyr and ZjQd, applying a standard quantizer in the context of an asynchrony-resistant FedAvg variant wouldn’t work directly.

In brief, the reason is that standard compressors (e.g. RandomK, QSGD) bound error with respect to the error of the vector being quantized. For FedAvg algorithms, this would either be the model/parameter, or the sum of gradient updates over several (asynchronous) rounds. The former wouldn’t work since the parameter can have an unbounded norm. For the latter, bounding the error under constant learning rate, as is the case for our algorithm, would require a gradient second-moment bound. (This fact is also reflected in e.g. the asynchronous analysis of [Koloskova et al., NeurIPS22], reference [A] by Reviewer ZjQd.)
We discuss this point in more detail in the answer to Reviewer oDyr.
This is why we adopted a different approach: we employ a non-trivial lattice quantizer, which bounds error with respect to a carefully-chosen “pivot” point. In turn, this allows us to handle both communication-quantization and asynchrony in the same analysis.

**To fully-address this issue, in the final revision we present experiments where we naively use the QSGD stochastic quantizer to compress updates both in QuAFL and in FedBuff, a state-of-the-art asynchronous FL algorithm. Our results in [Figure 5](https://openreview.net/pdf?id=Rb3RN0maB7F#page=8) and [Figure 16](https://openreview.net/pdf?id=Rb3RN0maB7F#page=16) show that, with very careful tuning of the learning rate, it is possible for the resulting algorithms to not diverge in this setting. However, we observe significantly worse convergence, given the much larger quantization error of these heuristics. We believe this justifies our use of the lattice quantizer.**

2. **Model of asynchrony:**

We support asynchrony arising from two sources:

* We *decouple* the client-server communication part of FedAvg, allowing the worker to perform updates on an “older” version of the model relative to the current one at the server

* Furthermore, we allow the server to directly read the current version of a worker’s model, before that version has received all the K local gradient updates.

We emphasize that both assumptions are very relaxed: we only assume that all clients are sampled uniformly by the server, and that each of them performs H > 0 steps *in expectation* at the point of contact. (Both assumptions are in fact necessary to avoid skew to the joint loss being optimized.) We do not assume a worst-case bound on delays.

**To fully address the reviewers' questions on this point, in the final revision we also compare against FedBuff, a state-of-the-art practical asynchronous FL algorithm. The results in [Figure 6](https://openreview.net/pdf?id=Rb3RN0maB7F#page=8) show that our algorithm outperforms FedBuff at the same number of local steps. Roughly, this is correlated to the fact that QuAFL takes into account partial progress by slow clients, whereas in FedBuff slow clients constantly contribute less significantly to the server updates, as they simply submit less updates to the buffer.**

3. **Additional questions:**

We also address the relationship to the concurrent work of [Koloskova et al., NeurIPS22], as well as questions about the experimental setup, in the responses to reviewers.

**We would be very happy if the reviewers would take the time to examine our revision, and would engage with us during the second discussion period.**

With best regards,

The QuAFL authors

---

### Author Response · Authors · 2022-12-11
**(Lack of) Feedback**

Dear Reviewers and ACs,

We must say that we are disappointed to not have received any feedback at any point during the discussion period.

This is despite the fact that we have provided:

1) Detailed responses to every point raised by the reviewers, including comparisons relative to concurrent work and to algorithms with different models and guarantees;

2) Clarifications for some of the non-trivial misunderstandings and inconsistencies which seeped into some of the reviews;

3) A revision addressing  each and every point raised, in particular containing all additional experiments asked for by the reviewers. For instance, we implemented existing quantizers in our setting, despite the fact that they do not have any convergence guarantees, showing that indeed they do not work as well as our approach even when tuned extremely carefully. Further, as suggested, we also implemented and compared against state-of-the-art asynchronous algorithms, showing that our algorithm indeed provides better practical performance.

Despite all of the above, our responses have remained unacknowledged throughout the discussion period. Needless to say, this is not what we were expecting when submitting to ICLR, which prides itself on active discussion. Moreover, with all due respect, we believe it is unfair to ask authors to perform significant additional work under deadline pressure without acknowledging it in any way during a months-long discussion period.

With best regards,

The authors

---

### Decision · Program_Chairs · 2023-01-20

**Decision:**

Reject

**Justification For Why Not Higher Score:**

The paper has issues on the overall novelty, the motivation, the writing, the significance of the theoretical results, and the correctness of the proofs.

**Justification For Why Not Lower Score:**

N/A

**Metareview: Summary, Strengths And Weaknesses:**

In this paper, the authors proposed QuAFL, which is a variant of federated averaging (i.e., FedAvg), by considering both the compression and asynchronous computations. Both topics have been extensively studied in the literature. Reviewers appreciate the authors' efforts on analyzing compression and  asynchrony together.  In the meanwhile, reviewers have many concerns on the overall novelty of the work, the writing of the paper, the motivation of the work, the significance of the theoretical results, the correctness of the analysis, and the thoroughness of the experiments. The rebuttals and additional experiments indeed helped reviewers (and improved the quality of the paper). However, the overall quality of the paper does not justify an acceptance. In particular, we feel the correctness of some of the analysis is still questionable even after reading the rebuttal, for example, see W1, W2, W3 from Reviewer ea7y . For future submissions, the authors are encouraged to invest significant effort to check the theoretical analysis to make sure the results are correct and the proofs are convincing.